# Connecting Large Language Models with Evolutionary Algorithms Yields Powerful Prompt Optimizers

**Qingyan Guo**[12†*], **Rui Wang**[2†], **Junliang Guo**[2], **Bei Li**[23], **Kaitao Song**[2], **Xu Tan**[2‡],
**Guoqing Liu**[2], **Jiang Bian**[2], **Yujiu Yang**[1‡]
[1]Tsinghua University    [2]Microsoft Research    [3]Northeastern University
gqy22@mails.tsinghua.edu.cn, libei_neu@outlook.com,
{ruiwa,junliangguo,kaitaosong,xuta,guoqingliu,jiabia}@microsoft.com
yang.yujiu@sz.tsinghua.edu.cn

## Abstract

Large Language Models (LLMs) excel in various tasks, but they rely on carefully crafted prompts that often demand substantial human effort. To automate this process, in this paper, we propose a novel framework for discrete prompt optimization, called EvoPrompt, which borrows the idea of evolutionary algorithms (EAs) as they exhibit good performance and fast convergence. To enable EAs to work on discrete prompts, which are natural language expressions that need to be coherent and human-readable, we connect LLMs with EAs. This approach allows us to simultaneously leverage the powerful language processing capabilities of LLMs and the efficient optimization performance of EAs. Specifically, abstaining from any gradients or parameters, EvoPrompt starts from a population of prompts and iteratively generates new prompts with LLMs based on the evolutionary operators, improving the population based on the development set. We optimize prompts for both closed- and open-source LLMs including GPT-3.5 and Alpaca, on 31 datasets covering language understanding, generation tasks, as well as BIG-Bench Hard (BBH) tasks. EvoPrompt significantly outperforms human-engineered prompts and existing methods for automatic prompt generation (e.g., up to 25% on BBH). Furthermore, EvoPrompt demonstrates that connecting LLMs with EAs creates synergies, which could inspire further research on the combination of LLMs and conventional algorithms. Our code is available at https://github.com/beeevita/EvoPrompt.

## 1 Introduction

Large language models (LLMs) show remarkable performance on multiple natural language processing (NLP) tasks (Touvron et al., 2023; Ouyang et al., 2022). To adapt to downstream tasks, simply adding an instruction to the input text, also called discrete prompt, steers LLMs to carry out the desired task with negligible impact on computational cost (Liu et al., 2023). Such approach also eliminates the need for all the parameters and gradients in LLMs, making it suitable for LLMs with block-box APIs such as GPT-3 and GPT-4 (Brown et al., 2020; OpenAI, 2023). Despite the convenience, the performance of the LLMs towards a certain task is significantly influenced by the prompt (Liu et al., 2023; Zhu et al., 2023). Accordingly, the key challenge of this approach lies in the design of the prompt, which has emerged as a crucial technique known as prompt engineering (Zhou et al., 2022). Given the wide variation in prompts across language models and tasks, the prompt design typically requires substantial human effort and expertise with subjective and relatively limited guidelines (Mishra et al., 2022a;b; Liu et al., 2023; Zamfirescu-Pereira et al., 2023; Wang et al., 2023).

---

*Work done during an internship at Microsoft Research Asia.
†Equal Contribution.
‡Corresponding Author.

To alleviate human effort on discrete prompt design, previous approaches usually rely on access to the token probabilities from the output layer of LLMs, which may not always be accessible through APIs (Deng et al., 2022; Zhang et al., 2023a). Some recent works consider enumerating diverse prompts and selecting the best ones (Zhou et al., 2022; Jiang et al., 2020), or modifying current prompts to improve them (Guo et al., 2023; Prasad et al., 2022; Pryzant et al., 2023). Such approaches either emphasize *exploring* diverse prompts, which may lead to indecisiveness and wasted resources, or focus on *exploiting* upon the current identified good prompts, which may result in stagnation and confine the search to local optima. Several conventional derivative-free algorithms are well-designed and strike a good balance between *exploration* and *exploitation* (Conn et al., 2009; Rios & Sahinidis, 2013). Among these, evolutionary algorithms (EAs) stand out as they are simple and efficient, as well as suitable for discrete prompt optimization (Storn & Price, 1997; Brest et al., 2006; Zhang & Sanderson, 2009; Vesterstrom & Thomsen, 2004). Sequences of phrases in prompts can be regarded as gene sequences in typical EAs, making them compatible with the natural evolutionary process.

In this paper, we borrow the idea of EAs and propose a discrete prompt tuning framework, EVO-PROMPT. While evolutionary operators in EAs are typically designed for sequences, they tend to independently alter tokens to generate new candidate solutions. Unfortunately, this approach ignores the connections among tokens, which is crucial for maintaining coherence and readability in prompts. Taking advantage of LLMs' expertise in NLP and the exceptional optimization capabilities of EAs, we connect these two approaches, where LLMs generate new candidate prompts following evolutionary operators, and EAs guide the optimization process to retain the optimal prompts.

Specifically, based on several initial prompts, we utilize LLMs to act as evolutionary operators to generate new prompt candidates, and the prompt with better performance on the development set is preserved. The above operations upon the updating population are iteratively applied to improve the quality. By elaborately designing the evolutionary operators and adjusting the update strategy, EVOPROMPT can be instantiated with various types of EAs. We optimize the prompts for two different LLMs (i.e., Alpaca (Taori et al., 2023), and GPT-3.5 (Brown et al., 2020)) on a diverse range of neural language understanding and generation tasks, as well as challenging BIG-Bench tasks, using a total of 31 datasets. EVOPROMPT consistently gets better prompts compared with both manually designed ones and previous automatic prompt generation methods. The main contributions of this paper include:

- We propose a novel framework for automatic discrete prompt optimization connecting LLMs and EAs, called EVOPROMPT, which enjoys the following advantages: **1)** It does not require access to any parameters or gradients of LLMs; **2)** It strikes a balance between *exploration* and *exploitation* leading to better results; **3)** The generated prompts are human-readable.

- Experiments conducted on 31 datasets demonstrate the effectiveness of EVOPROMPT compared with crafted prompts, as well as existing methods. We release the optimal prompts obtained by EVOPROMPT for these common tasks such as sentiment classification, topic classification, subjectivity classification, simplification, summarization and reasoning.

- We demonstrate that LLMs are capable of implementing multiple types of EAs provided with appropriate instructions. We hope that our explorations will inspire further investigations on the combination of LLMs and conventional algorithms, paving the way for new and innovative applications of LLMs.

## 2 RELATED WORKS

**Prompts in LLMs**   Prompting is an efficient method for employing LLMs in specialized tasks. However, the performance is heavily influenced by the choice of the prompt. Recently, automatic prompt optimization has obtained wide attention. Continuous prompt-based methods, which only tune parameters of some input tokens (Li & Liang, 2021; Liu et al., 2021b;a; Zhang et al., 2021) attract lots of attention. In spite of their effective performance, two drawbacks of such paradigms can not be ignored: **1)** The optimization of continuous prompts requires parameters of LLMs that are inaccessible for black-box APIs. **2)** Soft prompts often fall short of interpretability (Lester et al., 2021). Discrete prompts, simply adding several discrete tokens, such as "It was" (Schick & Schütze, 2021), or task-specific descriptive instructions, such as "Classify the comment into positive or negative.", to the input text, can offer an interactive interface to humans with better interpretability and show promising performance in various NLP tasks (Liu et al., 2023).

**Discrete Prompts**   Various approaches have been proposed for automatic discrete prompt searching and generation (Shin et al., 2020; Shi et al., 2022; Wallace et al., 2019; Deng et al., 2022; Zhang et al., 2023a), while these methods still rely on the gradients or the token probabilities from the output layer. More recently, considering the high variance of different prompts for downstream tasks, some works focus on *exploration* by enumerating and selecting the best prompt from a number of candidates, mainly augmented by re-sampling (Zhou et al., 2022; Jiang et al., 2020). Approaches based on prompt edit (Zhang et al., 2023a; Prasad et al., 2022) emphasize *exploitation*, which may potentially lead to local optima. Another approach collects the incorrectly predicted cases and analyzes the corresponding root cause to improve existing prompts (Pryzant et al., 2023; Guo et al., 2023), which also emphasizes *exploitation*. Additionally, such approaches are constrained to tasks with standard answers and cannot be directly applied to generation tasks. Our proposed EVOPROMPT empowered with evolutionary algorithms strikes a balance between *exploration* and *exploitation* without requiring any parameters or gradients.

**LLMs and Optimization Algorithms**   LLMs demonstrate the potential to serve as black-box optimizers (Zheng et al., 2023); however, this black-box approach lacks explainability. Some works have revealed that LLMs have the capability to imitate specific operations in conventional algorithms. For instance, LLMs can perform "Gradient Descent" in discrete space by collecting incorrectly predicted samples (Pryzant et al., 2023; Guo et al., 2023). Meanwhile, it has been demonstrated that LLMs can imitate the mutation (Lehman et al., 2022) or crossover (Meyerson et al., 2023) operator in the genetic algorithm (GA). Chen et al. (2023) further integrates LLMs and GA for neural architecture search, while Lanzi & Loiacono (2023) introduce a similar approach to game design. Our work has taken a significant step forward by proposing a general framework that connects LLMs with evolutionary algorithms, which can be instantiated to a diverse range of evolutionary algorithms through customization of evolutionary and selection processes, thereby broadening its applicability and potential influence in the domain. We aspire this work to inspire broader applications of combining LLMs and conventional algorithms.

## 3   AUTOMATIC DISCRETE PROMPT OPTIMIZATION

---

**Algorithm 1** Discrete prompt optimization: EVOPROMPT

---

**Require:** Initial prompts $P_0 = \{p_1, p_2, \ldots, p_N\}$, size of population $N$, a dev set $\mathcal{D}$, $f_{\mathcal{D}}(\cdot)$ denotes the score of a prompt on the desired LLM evaluated on $\mathcal{D}$, a pre-defined number of iterations $T$, carefully designed evolutionary operators to generate a new prompt $\text{Evo}(\cdot)$

1: **Initial evaluation scores**: $S_0 \leftarrow \{s_i = f_{\mathcal{D}}(p_i) | i \in [1, N]\}$
2: **for** $t = 1$ to $T$ **do**
3:     **Selection**: select a certain number of prompts from current population as parent prompts $p_{r_1}, \ldots, p_{r_k} \sim P_{t-1}$
4:     **Evolution**: generate a new prompt based on the selected parent prompts by leveraging LLM to perform evolutionary operators $p_i' \leftarrow \text{Evo}(p_{r_1}, \ldots, p_{r_k})$
5:     **Evaluation**: $s_i' \leftarrow f(p_i', \mathcal{D})$
6:     **Update**: $P_t \leftarrow \{P_{t-1}, p_i'\}$ and $S_t \leftarrow \{S_{t-1}, s_i'\}$ based on the evaluation scores
7: **end for**
8: **Return** the best prompt, $p^*$, among the final population $P_T$: $p^* \leftarrow argmax_{p \in P_T} f(p, \mathcal{D})$

---

Current advanced LLMs are typically interacted via black-box APIs, while the gradients and parameters are inaccessible. Evolutionary algorithms (EAs) are derivative-free algorithms with exceptional accuracy and rapid convergence. Accordingly, we consider introducing EAs into discrete prompt optimization. However, to generate new candidate solutions, evolutionary operators typically edit the elements in current solutions independently, without considering the connections between them. This makes it challenging to apply evolutionary operators on discrete prompts, which require coherence and readability. To address this challenge, we propose a synergistic approach that connects the natural language processing expertise of LLMs with the optimization capabilities of EAs, called EVOPROMPT. Specifically, LLMs generate new candidate prompts based on evolutionary operators, while EAs guide the optimization process to find the optimal prompts.

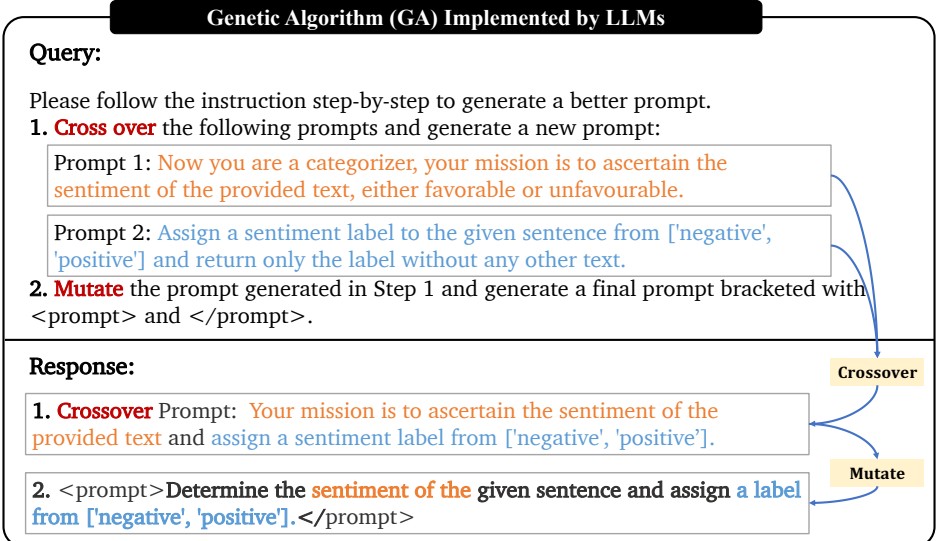

Figure 1: GA process implemented by LLMs (Evo(·) in Algorithm 1). In Step 1, LLMs perform *crossover* on the given two prompts (words in orange and blue are inherited from Prompt 1 and Prompt 2, respectively). In Step 2, LLMs perform *mutation* on the prompt.

In order to implement EVOPROMPT in practice, it is necessary to instantiate it with a specific algorithm of EAs. There are various types of EAs, and in this paper, we consider two widely used algorithms, including Genetic Algorithm (GA) (Holland, 1975) and Differential Evolution (DE) (Storn & Price, 1997). GA is among the most highly regarded evolutionary algorithms (Holland, 1975; 1992; Mitchell, 1998; Mirjalili et al., 2020) and DE has emerged as one of the most widely utilized algorithms for complex optimization challenges since its inception (Storn & Price, 1997; Price, 2013; Das & Suganthan, 2010; Pant et al., 2020). In the following, we will first outline the proposed EVOPROMPT, and then instantiate EVOPROMPT with GA and DE respectively.

## 3.1 FRAMEWORK OF EVOPROMPT

EAs typically start with an initial population of $N$ solutions (prompts in our setting), then iteratively generate new solutions using evolutionary operators (e.g., mutation and crossover) on the current population and update it based on a fitness function. Following typical EAs, EVOPROMPT mainly contains three steps:

- **Initial population**: Contrary to most existing automatic prompt methods that neglect priori human knowledge, we apply available manual prompts as the initial population to leverage the wisdom of humans. Besides, EAs typically start from random solutions, resulting in a diverse population and avoiding being trapped in a local optimum. Accordingly, we also introduce some prompts generated by LLMs (Zhou et al., 2022) into the initial population.

- **Evolution**: In each iteration, EVOPROMPT uses LLMs as evolutionary operators to generate a new prompt based on several parent prompts selected from the current population. To accomplish this, we design steps of the *mutation* and *crossover* operators for each specific type of EAs, along with corresponding instructions to guide the LLMs in generating new prompts based on these steps.

- **Update**: We evaluate the generated candidate prompts on a development set and retain those with superior performance, similar to the survival of the fittest in nature. The specific updating strategy may vary depending on the type of EAs used.

The algorithm stops when the number of iterations reaches a predefined value. The details of EVOPROMPT are outlined in Algorithm 1. When instantiating EVOPROMPT with a specific algorithm of EAs, the evolutionary processes need to be adjusted, and the key challenge is to design the evolutionary operators on discrete prompts.

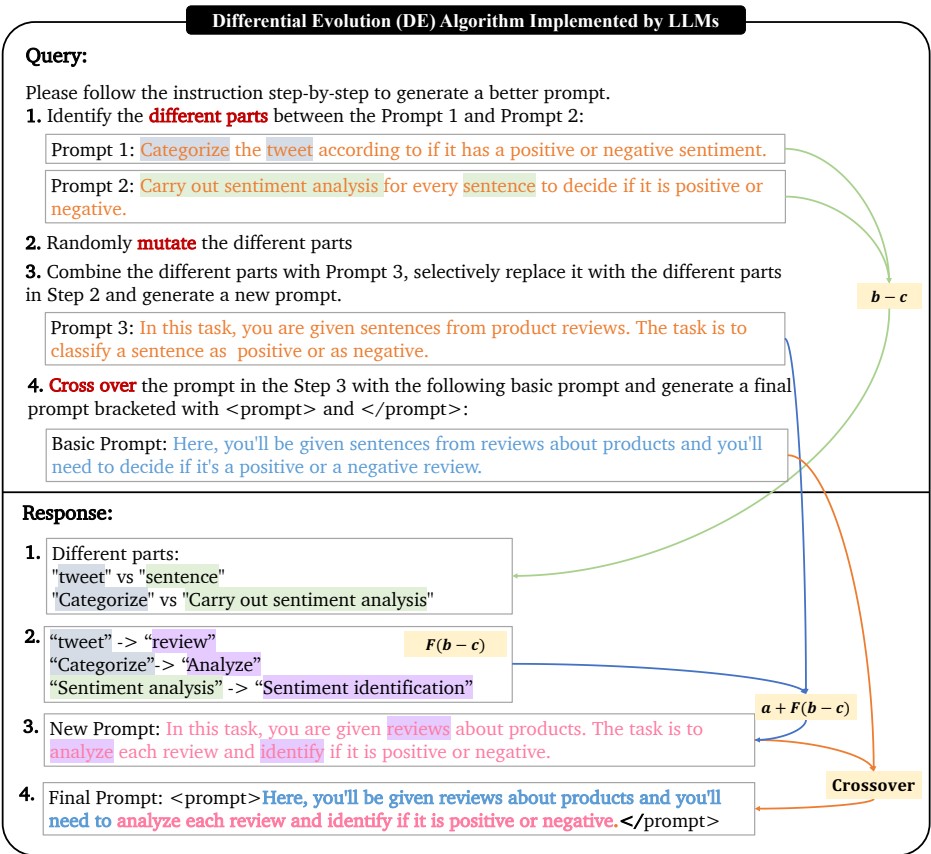

Figure 2: DE process implemented by LLMs (Evo(·) in Algorithm 1). In Step 1, LLMs find the different parts (words in ▆ and ▆) between Prompt 1 and Prompt 2 ($\mathbf{b} - \mathbf{c}$ in typical DE). In Step 2, LLMs perform *mutation* (words in ▆) on them (imitation of $\mathbf{F}(\mathbf{b} - \mathbf{c})$). Next, LLMs incorporate the current best prompt as Prompt 3 with the mutated results in Step 2, to generate a new prompt (counterpart of $\mathbf{a} + \mathbf{F}(\mathbf{b} - \mathbf{c})$ in DE). Finally, LLMs perform *crossover* upon the current basic prompt $p_i$ and the generated prompt in Step 3. See Figure 5 in Appendix B.2 for the complete response.

## 3.2 INSTANTIATION WITH GENETIC ALGORITHM

**Selection** In GA, parent solutions are conventionally selected using the roulette wheel selection method, guided by their fitness values (Lipowski & Lipowska, 2012). Analogously, we employ the roulette wheel selection to choose two parent prompts from the current population, based on their performance scores obtained on the development sets. Let $s_i$ denote the performance score of the $i$-th prompt within a population containing $N$ prompts. The probability of selecting the $i$-th prompt as a parent can be expressed as $p_i = s_i / \sum_{j=1}^{N} s_j$.

**Evolution** Conforming to the GA framework, we generate a new candidate prompt via two steps: **1)** Crossover is performed between the parent prompts to produce a new offspring prompt that inherits characteristics from both parents; **2)** Mutation is applied to the offspring prompt, introducing random alterations to certain elements. We formalize this two-stage operation into algorithmic instructions for guiding LLMs to implement Evo(·) in Algorithm 1. The entire process is illustrated in Figure 1.

**Update** We employ a straightforward selection strategy for updating the population: at each iteration, EVOPROMPT produces $N$ new prompts, which are merged with the existing population of $N$ prompts. Subsequently, the top $N$ prompts, based on their scores, are retained to form the updated population. Accordingly, the overall quality of the population undergoes continuous enhancement, culminating in the selection of the best one within the final population as the optimal prompt.

### 3.3 Instantiation with Differential Evolution

Here, we begin with some preliminary knowledge of DE. Unlike GA, the solutions of DE are represented by numerical vectors. Each vector within the population is sequentially selected as a base vector, denoted as $\mathbf{x}$, which subsequently undergoes mutation and crossover. During mutation, a mutated solution $\mathbf{y}$ is generated from a randomly selected solution $\mathbf{a}$ from the current population. The mutation is achieved by adding a scaled difference between two distinct, randomly selected solutions $\mathbf{b}$ and $\mathbf{c}$ to $\mathbf{a}$, i.e., $\mathbf{y} = \mathbf{a} + F(\mathbf{b} - \mathbf{c})$, where $F$ is the scaled parameter.

Crossover is to generate a trial solution $\mathbf{x}' = [x'_1, ..., x'_n]$ by choosing each parameter in the vector from either the basic solution $\mathbf{x}$ or the mutated solution $\mathbf{y}$. Then, $\mathbf{x}$ is replaced with $\mathbf{x}'$ if $\mathbf{x}'$ is better than $\mathbf{x}$. Within step-by-step evolution, DE ends with a population of high quality. A modified version of DE uses the current best solution as vector $\mathbf{a}$ to exploit information from the best one.

**Evolution**  The evolutionary process of DE can be decoupled into three steps: 1) $F(\mathbf{b} - \mathbf{c})$; 2) $\mathbf{y} = \mathbf{a} + F(\mathbf{b} - \mathbf{c})$; 3) Crossover of $\mathbf{x}$ and $\mathbf{y}$. In EVOPROMPT based on DE, we follow the three steps to design the evolutionary process, as well as the corresponding instructions for LLMs to generate a new prompt based on these steps as illustrated in Figure 2:

- Inspired by the differential vector in DE, we consider mutating only the different parts of two randomly selected prompts in the current population (Step 1 and Step 2 in Figure 2). The prompts in the current population are considered the current best ones. Accordingly, the shared components of two prompts tend to have a positive impact on the performance, and thus need to be preserved.

- A variant of DE employs the current best vector during the mutation process, where a mutated vector is generated by adding the scale of the differential vector to the current best vector. Building upon this idea, we generate a mutated prompt by selectively replacing parts of the current best one with the mutated different parts for combination. (Step 3 in Figure 2).

- Crossover replaces certain components of a basic prompt (i.e., a candidate of the current population) with segments from the mutated prompt. This operation combines the features of two different prompts, potentially creating a new and improved solution (Step 4 in Figure 2).

**Update**  Following the standard DE, each prompt $p_i$ in the current population is chosen as a basic prompt in turn to generate a corresponding new prompt $p'_i$ using the instruction in Figure 2. Then, the prompt with a higher score, either $p_i$ or $p'_i$, is retained. Accordingly, the population size remains constant while the overall quality of the population is enhanced.

## 4 Experiments

### 4.1 Implementation Details and Baselines

With GPT-3.5 performing evolutionary operators, we optimize prompts using EVOPROMPT for the open-source Alpaca-7b (Taori et al., 2023) and closed-source GPT-3.5 (text-davinci-003) (Brown et al., 2020). We pick the prompt with the highest score on the development set and report its score on the test set. Results reported on Alpaca are averaged over 3 random seeds and the standard deviation is provided, while for GPT-3.5, we report results of one seed due to budget limitation. In our evaluation, we compare EVOPROMPT against three categories of prompt-based approaches, detailed as follows:

- **Manual Instructions (MI)**: These serve as task-specific guidelines and are crafted based on established works, specifically referenced from Zhang et al. (2023b) for language understanding, Sanh et al. (2021) for summarization, and Zhang et al. (2023c) for text simplification.

- **PromptSource** (Bach et al., 2022) and **Natural Instructions (NI)** (Mishra et al., 2022b): These repositories aggregate human-composed prompts across a diverse range of datasets.

- **APE** (Zhou et al., 2022) and **APO** (Pryzant et al., 2023): APE employs an iterative Monte Carlo Search strategy, emphasizing on *exploration*. We reproduce it and initialize populations of equivalent sizes to that of EVOPROMPT. APO harnesses incorrectly predicted instances as "pseudo-gradient" to iteratively refine the original prompt, which emphasizes *exploitation*. We reproduce APO on binary classification tasks with the optimal manual prompt as the initial one.

| Method | SST-2 | CR | MR | SST-5 | AG's News | TREC | Subj | Avg. |
|---|---|---|---|---|---|---|---|---|
| MI (Zhang et al., 2023b) | 93.68 | **91.40** | 88.75 | 42.90 | 70.63 | 50.60 | 49.75 | 71.07 |
| NI (Mishra et al., 2022c) | 92.86 | 90.90 | 89.60 | 48.64 | 48.89 | 55.00 | 52.55 | 68.21 |
| PromptSource (Bach et al., 2022) | 93.03 | - | - | - | 45.43 | 36.20 | - | - |
| APE (Zhou et al., 2022) | 93.45(0.14) | 91.13(0.45) | 89.98(0.29) | 46.32(0.49) | 71.76(2.81) | 58.73(1.37) | 64.18(0.59) | 73.80 |
| APO (Pryzant et al., 2023) | 93.87(0.39) | 91.20(0.04) | 89.85(0.35) | - | - | - | 70.55(1.02) | - |
| EVOPROMPT (GA) | **95.13**(0.21) | 91.27(0.06) | 90.07(0.25) | **49.91**(0.61) | 72.81(0.61) | **64.00**(0.16) | 70.55(2.58) | 76.25 |
| EVOPROMPT (DE) | 94.75(0.21) | **91.40**(0.04) | **90.22**(0.09) | 49.89(1.73) | **73.82**(0.35) | 63.73(1.54) | **75.55**(2.26) | **77.05** |

Table 1: Main results on language understanding (accuracy) on Alpaca-7b.

| Method | Alpaca | | | GPT-3.5 | | |
|---|---|---|---|---|---|---|
| | ROUGE-1 | ROUGE-2 | ROUGE-L | ROUGE-1 | ROUGE-2 | ROUGE-L |
| MI (Sanh et al., 2021) | 35.92 | 11.16 | 31.67 | 43.95 | 17.11 | 39.09 |
| APE (Zhou et al., 2022) | 35.44(0.79) | 10.60(0.38) | 31.80(0.50) | 43.43 | 16.72 | 38.25 |
| EVOPROMPT (GA) | 38.46(1.45) | 13.36(0.75) | 34.20(1.40) | 45.22 | 18.52 | 41.06 |
| EVOPROMPT (DE) | **39.46**(0.51) | **13.93**(0.33) | **35.49**(0.56) | **46.49** | **19.49** | **41.96** |

Table 2: Main results on SAMSum dataset (summarization task) for Alpaca-7b and GPT-3.5.

## 4.2 LANGUAGE UNDERSTANDING

**Datasets and Settings**   We first conduct experiments on language understanding tasks across 7 datasets to validate our methods, including sentiment classification (SST-2 (Socher et al., 2013), MR (PANG, 2005), CR (Hu & Liu, 2004), SST-5 (Socher et al., 2013)), topic classification (AG's News (Zhang et al., 2015), TREC (Voorhees & Tice, 2000)) and subjectivity classification (Subj (Pang & Lee, 2004)). To constrain the output label space, we prepend the demonstration consisting of one example per class before the test case. See Appendix B for more details.

**Main Results**   Table 1, shows that: **1)** Compared with previous works on prompt generation and human written instructions, EVOPROMPT based on both GA and DE delivers significantly better results. **2)** EVOPROMPT (GA) is slightly better than EVOPROMPT (DE) on sentiment classification datasets. When it comes to topic classification datasets, EVOPROMPT (DE) performs better. Notably, on the subjectivity classification task (Subj), EVOPROMPT (DE) exhibits a substantial improvement over its GA counterpart, achieving a 5% accuracy advantage. This may be contributed by the exceptional ability of DE to evade local optima when the initial prompts are not of high quality.

## 4.3 LANGUAGE GENERATION

**Datasets and Settings**   For language generation, we evaluate our EVOPROMPT on text summarization and simplification tasks. For summarization, we adopt SAMSum (Gliwa et al., 2019), a challenging and intricate dialogue summarization dataset, and report ROUGE-1/2/L scores on Alpaca-7b and GPT-3.5. For text simplification, which aims to simplify the source text while preserving its original meaning, we employ the ASSET dataset (Alva-Manchego et al., 2020), a

| Method | Alpaca | GPT-3.5 |
|---|---|---|
| MI (Zhang et al., 2023c) | 43.03 | 43.80 |
| APE (Zhou et al., 2022) | 45.90(0.09) | 46.71 |
| EVOPROMPT (GA) | 46.43(0.19) | 47.36 |
| EVOPROMPT (DE) | 46.21(0.27) | 47.40 |

Table 3: Main results (SARI) on simplification (ASSET) for Alpaca-7b and GPT3.5.

benchmark known for its multiple reference translations. We apply SARI score (Xu et al., 2016) as the evaluation metric, an n-gram-based scoring system extensively utilized for text editing tasks. Additional details regarding our experimental setup can be found in Appendix B.

**Main Results**   The summarization and simplification results are presented in Tables 2 and 3. EVOPROMPT achieves a substantial performance gain over manually designed prompts, exhibiting an improvement of over 3 points in SARI scores across both Alpaca and GPT-3.5 API. Furthermore, EVOPROMPT consistently outperforms the APE approach across the evaluated scenarios, indicating

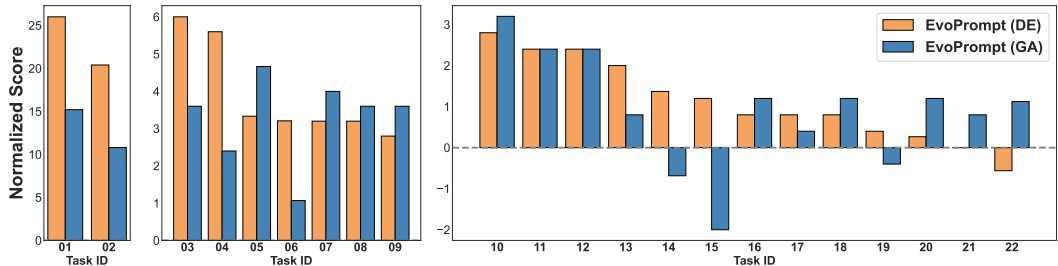

Figure 3: Normalized scores on BBH tasks for EVOPROMPT (GA) and EVOPROMPT (DE).

that the generated prompts effectively harness the capabilities of LLMs for superior performance. Moreover, EVOPROMPT (DE) notably outperforms EVOPROMPT (GA) in the summarization task, while demonstrating comparable performance in the text simplification task. This suggests that the DE variant is particularly effective for more complex language generation tasks like summarization.

### 4.4 BIG BENCH HARD (BBH)

**Datasets and Settings** To validate our methods on diverse tasks, we apply BBH (Suzgun et al., 2022) including a suite of 23 challenging BIG-Bench tasks requiring multi-step reasoning. Since these tasks are challenging, we focus on optimizing the prompts for GPT-3.5. We sample a subset from the test set as the development set and report the normalized scores[1] in comparison to the prompt "Let's think step by step." (Kojima et al., 2022) with 3-shot Chain-of-Thought demonstrations (following Fu et al. (2023)) on the test set. We use task IDs to simplify the denotation of each task and remove one since the accuracy already reaches 100% with the manual prompt. Please see Appendix C.2 and Table 17 for details, as well as further comparisons with previous works.

**Main Results** EVOPROMPT obtains better prompts for all 22 tasks (Figure 3). Specifically, EVO-PROMPT (DE) achieves up to a 25% improvement with an average of 3.5%, whereas EVOPROMPT (GA) reaches a peak improvement of 15% with a 2.5% average. Though for some tasks the GA counterpart outperforms the DE version, the performance gap remains relatively small (i.e., around 1%). Meanwhile, EVOPROMPT (DE) surpasses EVOPROMPT (GA) by over 2% on 6 tasks. Accordingly, the DE version is generally a good choice for these challenging tasks.

## 5 ANALYSIS

### 5.1 DESIGNS IN GA

For EVOPROMPT (GA), we apply the roulette wheel selection strategy by default to select parental prompts, contributing to the offspring. To further explore the effect of various selection strategies, we compare our approach with another two popular strategies, i.e., tournament (Wikipedia contributors, 2023) and random selection, as presented in Table 4. We observe that EVOPROMPT (GA) with roulette wheel achieves higher scores, showcasing the effectiveness of this selection method.

| Strategy | SST-5 | ASSET | Avg. |
|---|---|---|---|
| random | 48.67(0.97) | 46.32(0.32) | 47.50 |
| tournament | 49.70(0.60) | 46.29(0.18) | 48.00 |
| wheel | **49.91**(0.61) | **46.43**(0.19) | **48.17** |

Table 4: Designs in EVOPROMPT (GA).

### 5.2 DESIGNS IN DE

For EVOPROMPT (DE), we delve into two key design considerations in adapting the evolutionary operators of DE to discrete prompts: **1)** mutation on different parts, and **2)** choosing the current top-performing prompt as "Prompt 3" in Figure 2. We assess the impact of these design choices on

---

[1]The accuracy difference between a given prompt and the baseline prompt "Let's think step by step." A score of 0 corresponds to the normalized score of the baseline prompt.

two datasets: Subj, an understanding dataset where EVOPROMPT (DE) outperforms EVOPROMPT (GA), and ASSET, a generation dataset where both variants demonstrate similar performance.

**Mutation on Different Parts**    To illustrate the benefits of mutating only the different parts, we replace the first two steps in Figure 2 with the instruction "Randomly mutate Prompt 1 and Prompt 2" to allow mutation on all contents in Prompts 1 and 2, denoted as "All" in Table 5. Meanwhile, the original design in EVOPROMPT, which mutates only the different parts, is denoted as "Diff". As shown in Table 5, the design of mutation on only the different parts consistently yields performance gains across two tasks.

| Mutation | Prompt 3 | Subj | ASSET |
|----------|----------|------|-------|
| Diff | best | **75.55**(2.26) | **46.21**(0.27) |
| All | best | 69.87(0.82) | 45.73(0.45) |
| Diff | random | 69.82(2.47) | 45.89(0.37) |
| Diff | eliminate | 69.07(4.21) | 45.90(0.23) |

Table 5: Designs in EVOPROMPT (DE).

**Selection of Prompt 3**    Applying one of the variants of the DE algorithm, in EVOPROMPT (DE), we pick the best prompt in the current population as Prompt 3 in Figure 2. We validate this design via the following settings: **1)** Prompt 3 is randomly sampled from the current population, denoted as "random" in Table 5; **2)** Eliminate the use of Prompt 3 by letting the Basic Prompt directly cross over with the mutated different parts (i.e., remove Step 3 in Figure 2), denoted as "eliminate" in Tabel 5. Table 5 clearly demonstrates the importance of introducing Prompt 3. Moreover, it is shown that choosing the best prompt as Prompt 3 is more effective than random sampling.

## 5.3    POPULATION INITIALIZATION

We investigate the effect of initial population quality on EVOPROMPT. We conduct pilot experiments to sort the prompts (designed manually or generated by GPT-3.5) according to their performance on the dev set. We then select bottom, random and top prompts along with their corresponding variations as initial prompts. These variations are generated using the resampling template designed in Zhou et al. (2022), shown in Figure 4 in the Appendix B.2, which is used to introduce randomness to the initialization.

| Initialization | GA | DE |
|----------------|-----|-----|
| bottom-10 | 47.80(0.92) | 48.64(0.15) |
| random-10 | 49.34(0.53) | **50.03**(1.08) |
| random-5 + var-5 | 49.84(1.49) | 49.53(1.04) |
| top-10 | 49.62(1.00) | 49.61(2.30) |
| top-5 + var-5 | **49.91**(0.61) | 49.89(1.73) |

Table 6: Ablations of the initial population on SST-5, where top-$n$, random-$n$, bottom-$n$ denotes the top-performing, randomly selected, bottom-performing n prompts, and var-$n$ denotes the number of generated $n$ variations.

Table 6 demonstrates that: **1)** Crafted design of initial prompts is not essential, as randomly selecting prompts can achieve a similar performance to selecting the top-performing ones; **2)** When selecting the top-performing prompts, introducing randomness by allowing GPT-3.5 to generate variations can lead to a slight improvement in overall performance; however, when randomly selecting prompts, there is no need to introduce additional randomness for EVOPROMPT (DE); **3)** When using top-performing initial prompts, EVOPROMPT (GA) performs slightly better than EVOPROMPT (DE); however, when starting with bottom-performing initial prompts, EVOPROMPT (DE) outperforms EVOPROMPT (GA), which indicates that DE is a better choice when the available manual prompts are not of high quality.

## 6    CONCLUSIONS

We introduce EVOPROMPT to optimize discrete prompts, which connects LLMs with evolutionary algorithms. Extensive experiments on 31 datasets demonstrate the superiority of EVOPROMPT, yielding consistent performance gains over both manual instructions and existing methods. Besides, We validate that LLMs can serve as an effective, interpretable interface for implementing evolutionary algorithms like GA and DE. While this study focused on EAs, the extensibility of our approach opens avenues for applying LLMs to other conventional algorithms, such as particle swarm optimization (PSO) (Kennedy & Eberhart, 1995), ant colony optimization (ACO) (Dorigo & Gambardella, 1997) and more recent Quality-Diversity (QD) optimization algorithms. Our findings aim to inspire future research at the intersection of LLMs and traditional algorithms, encouraging innovative applications.

ACKNOWLEDGEMENTS

This work was partly supported by the National Key Research and Development Program of China (No. 2020YFB1708200), and the Shenzhen Science and Technology Program (JCYJ20220818101001004).

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

---

**Algorithm 2** Discrete prompt optimization: EVOPROMPT (**GA**)

---

**Require:** Initial prompts $P_0 = \{p_1, p_2, \ldots, p_N\}$, size of population $N$, a dev set $\mathcal{D}$
1: **Initial fitness evaluation**: $S_0 \leftarrow \{s_i = f(p_i, D)|i \in [1, N]\}$
2: **for** $t = 1$ to $T$ **do**                    $\triangleright T$: Number of iterations
3:     **for** $i = 1$ to $N$ **do**
4:         **Selection** based on fitness using roulette wheel: $p_{r_1}, p_{r_2} \sim P_{t-1}$
5:         **Evolution**: $p_i' \leftarrow GA(p_{r_1}, p_{r_2})$ (Refer to Figure 1)
6:         **Evaluation**: $s_i \leftarrow f(p_i', \mathcal{D})$
7:     **end for**
8:     $S_t' \leftarrow \{s_i|i \in [1, N]\}, P_t' \leftarrow \{p_i'|i \in [1, N]\}$
9:     **Update score**: $S_t \leftarrow \text{Top-}N\{S_{t-1}, S_t'\}$
10:    **Update**: $P_t \leftarrow \text{Top-}N\{P_{t-1}, P_t'\}$ using $S_{t-1}, S_t'$,
11: **end for**
12: **Return** the best prompt, $p^*$, among the final population $P_T$: $p^* \leftarrow argmax_{p \in P_T} f(p, \mathcal{D})$

---

**Algorithm 3** Discrete prompt optimization: EVOPROMPT (**DE**)

---

**Require:** Initial prompts $P_0 = \{p_1, p_2, \ldots, p_N\}$, size of population $N$, a dev set $\mathcal{D}$
1: **for** $t = 1$ to $T$ **do**                    $\triangleright T$: Number of iterations
2:     **for** $p_i$ in $P_{t-1}$ **do**
3:         **Sample donors**: $p_{r1}, p_{r2} \sim P_{t-1}$ , $r1 \neq r2 \neq i$
4:         **Evolution**: $p_i' \leftarrow DE(p_i, p_{r_1}, p_{r_2}, p_{best})$ where $p_{best}$ is the current best prompt. (Refer to Figure 2)
5:         **Selection**: $p_i^* = \underset{p \in \{p_i, p_i'\}}{\arg\max} f(p, \mathcal{D})$           $\triangleright$ Keep the better one in the population
6:     **end for**
7:     **Update**: $P_t \leftarrow \{p_i^*|i \in [1, N]\}$
8: **end for**
9: **Return** the best prompt, $p^*$, among the final population $P_T$: $p^* \leftarrow argmax_{p \in P_T} f(p, \mathcal{D})$

---

## A DETAILS OF ALGORITHM IMPLEMENTATION

We instantiate EVOPROMPT two representative evolutionary algorithms, GA and DE. Though both algorithms use consistent general selection processes, creating offspring, and updating, it is worth noting that the selection strategies, ways of mutation and crossover, and the updating strategies in these two algorithms are different. The specific algorithms for each of them are shown in Algorithm 2 and Algorithm 3.

## B EXPERIMENTAL SETTINGS

### B.1 DATASETS

Table 7 shows the statistics of the text classification, simplification and summarization datasets. For Big-Bench Hard, We use serial numbers to denote 22 tasks, the descriptions are reported in Table 17. Note that for the task of "web of lies", the accuracy of the baseline is 100%, so here we have not included this task for prompt optimization. Additionally, both tasks of "logical deduction objects" and "tracking shuffled objects" have three sub-tasks.

### B.2 TEMPLATES

**Templates for Task Implementation** For different models, we apply different templates shown in Table 8, 9 and 10, referring to the previous works (Iyer et al., 2022; Taori et al., 2023; Zhang et al., 2023b; Li et al., 2023; Fu et al., 2023).

> **Template for Variation**
>
> Generate a variation of the following instruction while keep the semantic meaning.
> Input: <prompt>
> Output:

Figure 4: Template used for resampling (Zhou et al., 2022).

| Dataset | Type | Label space | \|Test\| |
|---------|------|-------------|-------|
| SST-2 | Sentiment | {positive, negative} | 1,821 |
| CR | Sentiment | {positive, negative} | 2,000 |
| MR | Sentiment | {positive, negative} | 2,000 |
| SST-5 | Sentiment | {terrible, bad, okay, good, great} | 2,210 |
| AG's News | News topic | {World, Sports, Business, Tech} | 7,600 |
| TREC | Question topic | {Description, Entity, Expression, Human, Location, Number} | 500 |
| Subj | Subjectivity | {subjective, objective} | 2,000 |
| SAMSum | Summarization | - | 819 |
| ASSET | Simplification | - | 359 |

Table 7: Statistics for natural language understanding and generation datasets used in this work.

**Template for Prompt Generation** We apply the resampling template, shown in Figure 4, to generate variations of manual initial prompts. For our EVOPROMPT, the complete DE algorithm implemented by LLMs is shown in Figure 5. For both DE and GA, we prepend a one-shot example of the algorithm execution, guiding LLMs to operate precisely.

---

============================ INSTRUCTIONAL PROMPTS ============================

Below is an instruction that describes a task, paired with an input that provides further context. Write a response that appropriately completes the request.

### Instruction:
<PROMPT>

### Input:
<INPUT>

### Response:
<COMPLETE>

---

**Zero-shot Example**:
Below is an instruction that describes a task, paired with an input that provides further context. Write a response that appropriately completes the request.

### Instruction:
Please perform Sentiment Classification task. Given the sentence, assign a sentiment label from ['negative', 'positive']. Return label only without any other text.

### Input:
beautifully observed , miraculously unsentimental comedy-drama .

### Response:
<COMPLETE>

---

Table 8: Template used for Alpaca (referring to Taori et al. (2023)).

## B.3 HYPER PARAMETERS

The parameters for the experiments are shown in Table 11. For evolutionary algorithms implemented by GPT-3.5, following previous work (Shi et al., 2024), we use Top-$p$ decoding (temperature=0.5, $P = 0.95$). For the task implementation, we use greedy decoding and the default temperature for Alpaca. For the generation tasks implemented by GPT-3.5, the temperature is $0.0$.

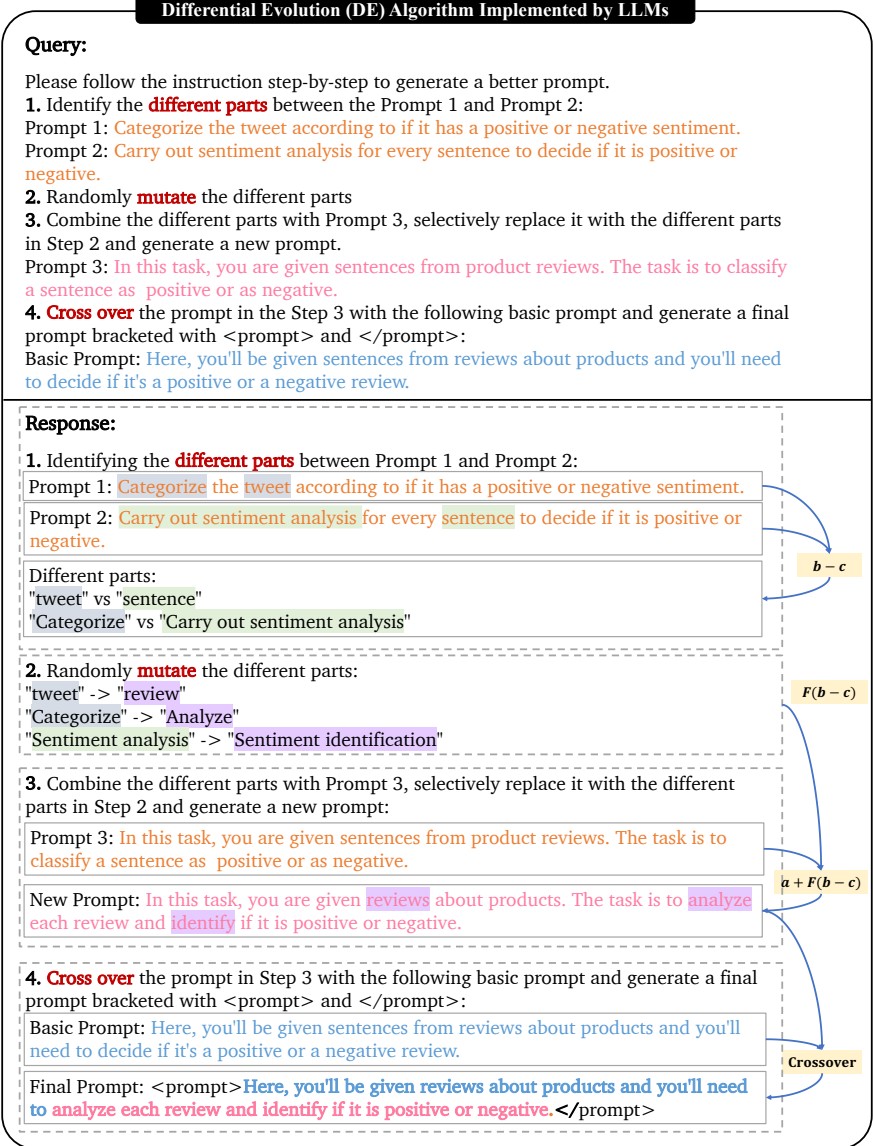

Figure 5: DE algorithm implemented by LLMs for discrete prompt optimization with **complete response** (Evo($\cdot$) in Algorithm 1). In Step 1, LLMs find the different parts (words in ▢ and ▢) between Prompt 1 and Prompt 2 ($\mathbf{b} - \mathbf{c}$ in typical DE). In Step 2, LLMs perform *mutation* (words in ▢) on them (imitation of $\mathbf{F}(\mathbf{b} - \mathbf{c})$). Next, LLMs incorporate the current best prompt as Prompt 3 with the mutated results in Step 2, to generate a new prompt (counterpart of $\mathbf{a} + \mathbf{F}(\mathbf{b} - \mathbf{c})$ in DE). Finally, LLMs perform *crossover* upon the current basic prompt $p_i$ and the generated prompt in Step 3.

```
========================= TEMPLATE FOR SIMPLIFICATION =========================

<PROMPT>
<INPUT>
The simplification of the sentence is <COMPLETE>

_______________________________________________________________________________

Zero-shot example:
Simplify the text.
Subsequently, in February 1941, 600 Jews were sent to Buchenwald and Mauthausen concentration camps.
The simplification of the sentence is <COMPLETE>

========================= TEMPLATE FOR SUMMARIZATION =========================

<PROMPT>
<INPUT>
TL;DR: <COMPLETE>

_______________________________________________________________________________

Zero-shot example:
How would you rephrase that in a few words?
Theresa: have you been at Tom's new place? Luis: yes, it's nice Marion: He invited us for a dinner Adam: where
is it? Marion: a bit outside the city Adam: where exactly? Marion: Fiesole Luis: very nice!
TL;DR: <COMPLETE>
```

Table 9: Templates of summarization (following Sanh et al. (2021); Qin et al. (2023)), simplification (following Li et al. (2023)) and the corresponding zero-shot examples.

```
==========================TEMPLATE FOR BIG-BENCH HARD =========================

<DESC>
Q: <INPUT>
A: <PROMPT>
<COMPLETE>

_______________________________________________________________________________

Zero-shot example:
Questions that involve enumerating objects and asking the model to count them.
Q: I have a flute, a piano, a trombone, four stoves, a violin, an accordion, a clarinet, a drum, two lamps, and a
trumpet. How many musical instruments do I have?
A: Let's think step by step.
<COMPLETE>
```

Table 10: Template for Big-Bench Hard (following Suzgun et al. (2022)) used for GPT-3.5 and the corresponding zero-shot examples. <DESC> refers to the specific description of each task.

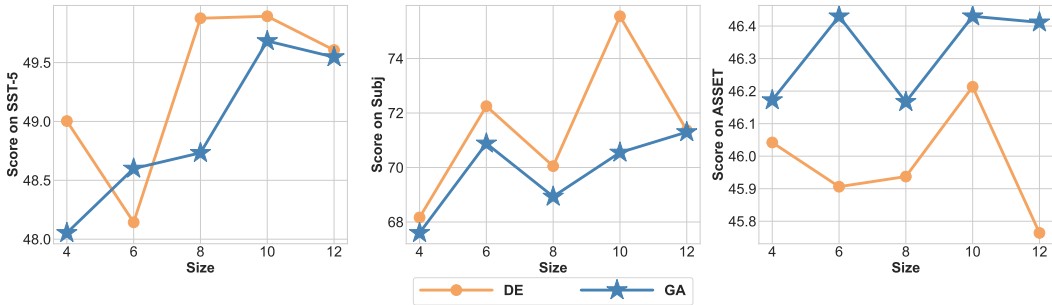

Figure 6: Effect of population size on SST-5 (left), Subj (middle), and ASSET (right). All the results are averaged over 3 random seeds.

**Text Classification** The population of prompts is initialized with widely used instructions in the previous works (Mishra et al., 2022b; Zhang et al., 2022). We paraphrase and rewrite them to initialize the population. The size of the development set is 200. We report the results on the full test set (the same as the previous related works (Deng et al., 2022; Zhang et al., 2023a)), as shown in Table 11.

**Text Generation** For the initial population, we collect instructions for summarization and simplification from Li et al. (2023); Sanh et al. (2021); Zhang et al. (2023c) and augment them to the expected size (10 in our setting), either written manually or generated by GPT-3.5.

| Task LM | ‖ | \|Population\| | \|Steps\| | \|Dev\| | \|Shots\| |
|---|---|---|---|---|---|
| *Text classification* | | | | | |
| Alpaca-7b | ‖ | 10 | 10 | 200 | 1 |
| *Text Generation* | | | | | |
| Alpaca-7b | ‖ | 10 | 10 | 100 | 0 |
| GPT-3.5 | ‖ | 10 | 10 | 100 | 0 |
| *Big-Bench Hard* | | | | | |
| GPT-3.5 | ‖ | 10 | 10 | 50 | 3 |

Table 11: Settings for experiments. \|Shots\| refers to the number of examples in the demonstration. For the text classification task, we set the value as 1, which means we prepend with 1 sample of each category, to constrain the output in the label space.

## C ADDITIONAL RESULTS

### C.1 PARAMETERS IN EVOLUTIONARY ALGORITHMS

**Effect of Population Size** Intuitively, a trade-off exists between the performance and the overhead caused by the population size. We explore the performance of EVOPROMPT (DE) and EVOPROMPT (GA) respectively at varying population sizes from 4 to 12. The results are plotted in Figure 6.

For classification datasets, as the size increases, curves for DE and GA show an ascending trend. Furthermore, the increase in DE attributed to population diversity was greater than that in GA since DE focuses on different parts. Differences among prompts within populations bring about substantial mutations, leading DE to explore potential prompts since keeping common parts balances exploration and exploitation effectively.

For the relatively simple generation task (i.e., ASSET), a population size of 6 demonstrates a comparable performance to a population size of 10, though with a 2.5-fold increase in overhead. This suggests that for relatively simple tasks large populations are unnecessary, while for complex tasks (i.e., Subj), a larger population with diversity brings improvement.

**Effect of Number of Iterations** To further explore the process of convergence, for SST-5, Subj and ASSET, we plot the best and average scores on the development set for EVOPROMPT for DE and GA over the whole population after each iterative step (Figure 7). Curves of best and average scores gradually converge with an increasing trend as evolution proceeds, indicating that the population's quality as a whole is steadily increasing as the evolution process.

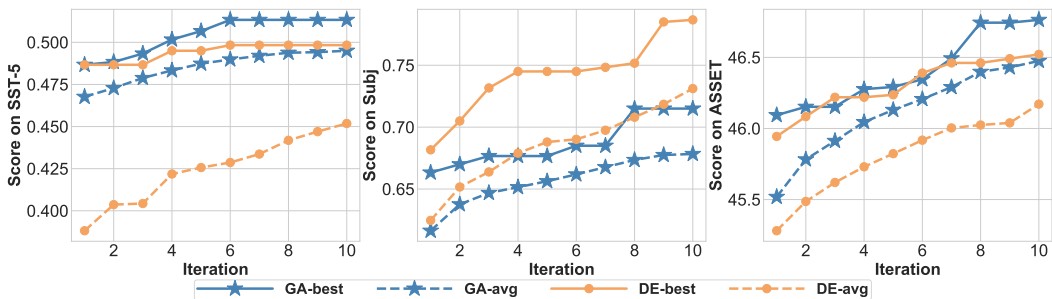

Figure 7: The best and average scores of each iteration on SST-5 (left), Subj (middle), and ASSET (right) development set on Alpaca-7b. All the results are averaged over 3 random seeds.

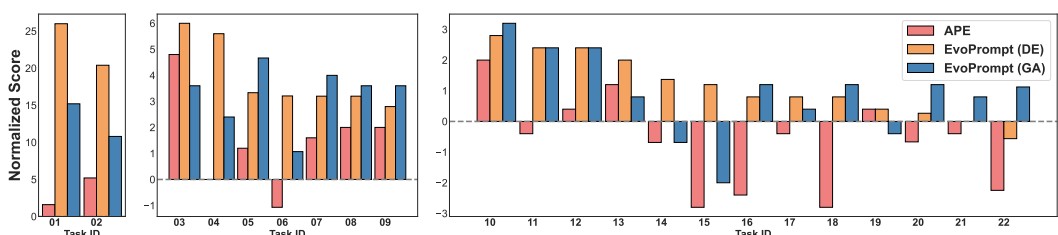

Figure 8: Normalized scores on BBH tasks for APE, EVOPROMPT (GA) and EVOPROMPT (DE).

## C.2 COMPARISON ON BBH TASKS

APE (Zhou et al., 2022) optimizes the Chain-of-Thought (CoT) prompt for reasoning tasks on InstructGPT. Considering that both InstructGPT and GPT-3.5 belong to the GPT family and we may observe similar trends, we evaluate the CoT prompt proposed by APE, "Let's work this out in a step by step way to be sure we have the right answer.", on reasoning tasks and plot the 3-shot performance in Figure 8. For simplicity, we use the same initial population for all the 22 BBH tasks without priori knowledge of each task. In future works, by incorporating task-specific prompts, either manually designed or generated by LLMs, we may further enhance the performance.

| Method | Avg. |
|---|---|
| baseline | 71.49 |
| APE | 71.85 |
| **EVOPROMPT (GA)** | 74.18 |
| **EVOPROMPT (DE)** | **75.03** |

Table 12: Average accuracy over 23 BBH tasks for different methods.

| | | SST-5 | | | Subj | |
|---|---|---|---|---|---|---|
| | APE | EVOPROMPT (GA) | EVOPROMPT (DE) | APE | EVOPROMPT (GA) | EVOPROMPT (DE) |
| | | | *Same iteration* | | | |
| # iterations | 9 | 9 | 9 | 15 | 15 | 15 |
| # tokens | 5.39 M | 5.40 M | 5.52 M | 5.66 M | 5.73 M | 5.93 M |
| score | 45.79 | 50.23 | 49.23 | 67.20 | 70.10 | 79.35 |
| | | | *Until convergence* | | | |
| # iterations | 9 | 7 | 11 | 15 | 15 | 17 |
| # tokens | 5.39 M | 4.20 M | 6.75 M | 5.66 M | 5.73 M | 6.72 M |
| score | 45.79 | 50.23 | 51.13 | 67.20 | 70.10 | 79.35 |

Table 13: Number of iterations, tokens within the API requests (including prompt optimization and evaluation) and the corresponding score for our methods and APE. We choose the iteration that APE converges as the *Same iteration* for comparison. *Until convergence* means that the improvement of the average score is less than 0.3% for continuous two iterations.

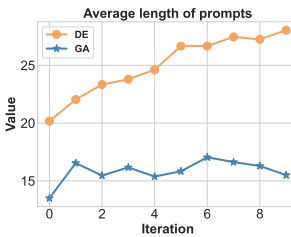 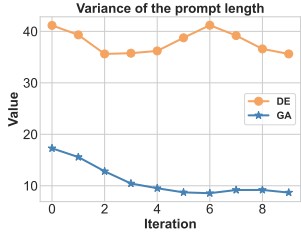 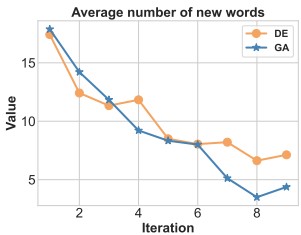

(a) Average length over the population after each step.

(b) Variance of prompt length over the population of each step.

(c) Number of new words generated after each step.

Figure 9: Statistics about the prompt length, including average values over the whole population (a), variance over the prompt length (b), and number of new words evolved after each step (c). Note that all the values are averaged over 8 datasets, including 7 understanding datasets and one simplification dataset, and 3 random seeds.

## C.3 COST ANALYSIS

Overhead mainly comes from prompt evaluation and generation. For evaluation, our overhead is $N * |D| * T$, where $N$ is the size of the population, $|D|$ is the size of the development set, and $T$ is the number of iterations. These parameters differ from the task and can be found in Appendix B.3. For the cost from prompt generation, the cost mainly depends on the number of API results, $T * N$. So the total number of API requests is $N * T * (1 + |D|)$, the same as APE. Moreover, given that the API of LLMs is typically billed based on the number of tokens used, we also estimate the total number of tokens used in the API requests during the prompt optimization process, as shown in Table 13. All the scores reported are over the test set on one random seed. We analyze the overhead mainly from two aspects: 1) the performance of our methods compared with APE under the *same number of iterations*; 2) the performance *until convergence* measured by the average score on the dev set.

We can observe that with the same number of iterations, both GA and DE outperform APE significantly while introducing only a slight overhead in terms of the number of tokens. The convergence rates of APE and GA are similar while DE is slightly slower, but it delivers better performance. This implies the relatively high ceiling of EVOPROMPT.

## C.4 ANALYSIS OF PROMPT

**Diversity Analysis** We further investigate the diversity of prompts generated by GA and DE after each iterative step respectively. We mainly plot the average prompt length, variance and number of new words mutated after each step, as shown in Figure 9. It can be observed that EVOPROMPT (DE) generates longer prompts with higher variances than EVOPROMPT (GA), which implies that DE prefers exploration for diversity. In the latter iterations, DE mutates more new words than GA, and thus shows better potential to escape from the local optimum.

**Optimal Prompts** We release the optimal prompts generated by EVOPROMPT for understanding (Table 14), text simplification (Table 16), summarization (Table 15) and BBH tasks (Table 17, 18) .

## D FUTURE WORKS

There are several promising directions for future investigation:

- Based on our framework, more applications can be explored, including game levels generation, text-to-images generation, non-trivial NP-hard problems (e.g. traveling salesman problem), etc.

- There exist many variants of DE and we give priority to the most canonical and classical ones for current exploration. In future work, it will be interesting to consider more advanced DE-variants (Das et al., 2016; Das & Suganthan, 2010). For example, some recent DE-variants have been investigating adaptive control parameters. The main challenge in applying these variants to

| Dataset | Method | Content | Score |
|---|---|---|---|
| SST-2 | Manual Instruction | Please perform Sentiment Classification task. Given the sentence, assign a sentiment label from ['negative', 'positive']. Return label only without any other text. | 93.68 |
| | Natural Instruction | In this task, you are given sentences from movie reviews. The task is to classify a sentence as "great" if the sentiment of the sentence is positive or as "terrible" if the sentiment of the sentence is negative. | 92.86 |
| | PromptSource | Does the following sentence have a positive or negative sentiment? | 93.03 |
| | **EVOPROMPT** | Examine the movie reviews and classify them as either positive or negative. | 95.61 |
| CR | Manual Instruction | Please perform Sentiment Classification task. Given the sentence, assign a sentiment label from ['negative', 'positive']. Return label only without any other text. | 91.40 |
| | Natural Instruction | In this task, you are given sentences from movie reviews. The task is to classify a sentence as "great" if the sentiment of the sentence is positive or as "terrible" if the sentiment of the sentence is negative. | 90.90 |
| | **EVOPROMPT** | Analyze customer reviews and categorize each sentence as either 'positive' or 'negative'. | 91.75 |
| MR | Manual Instruction | Please perform Sentiment Classification task. Given the sentence, assign a sentiment label from ['negative', 'positive']. Return label only without any other text. | 88.75 |
| | Natural Instruction | In this task, you are given sentences from movie reviews. The task is to classify a sentence as "great" if the sentiment of the sentence is positive or as "terrible" if the sentiment of the sentence is negative. | 89.60 |
| | **EVOPROMPT** | Identify if a movie review is positive or negative by accurately categorizing each input-output pair into either 'positive' or 'negative'. | 91.35 |
| SST-5 | Manual Instruction | Please perform Sentiment Classification task. Given the sentence, assign a sentiment label from ['terrible', 'bad', 'okay', 'good', 'great']. Return label only without any other text. | 42.90 |
| | Natural Instruction | In this task, you are given sentences from movie reviews. Based on the given review, classify it to one of the five classes: (1) terrible, (2) bad, (3) okay, (4) good, and (5) great. | 48.64 |
| | **EVOPROMPT** | Have your friend evaluate the movie they had just seen and provide a summary opinion (e.g. terrible, bad, okay, good, or great) to determine the sentiment of the movie review. | 52.26 |
| AG's News | Manual Instruction | Please perform News Classification task. Given the news item, assign a label from ['World', 'Sports', 'Business', 'Tech']. Return label only without any other text. | 70.63 |
| | Natural Instruction | In this task, you are given a news article. Your task is to classify the article to one out of the four topics "World", "Sports", "Business", "Tech" if the article's main topic is relevant to the world, sports, business, and technology, correspondingly. If you are not sure about the topic, choose the closest option. | 48.89 |
| | PromptSource | What label best describes this news article? | 45.43 |
| | **EVOPROMPT** | Assess the entire concept of the news story and choose from the World, Sports, Business or Tech categories to categorize it into the correct category. | 76.21 |
| TREC | Manual Instruction | Please perform Question Classification task. Given the question, assign a label from ['Description', 'Entity', 'Expression', 'Human', 'Location', 'Number']. Return label only without any other text. | 50.60 |
| | Natural Instruction | You are given a question. You need to detect which category better describes the question. Answer with "Description", "Entity", "Expression", "Human", "Location", and "Number". | 55.00 |
| | PromptSource | Which category best describes the following question? Choose from the following list: Description, Entity, Abbreviation, Person, Quantity, Location. | 36.20 |
| | **EVOPROMPT** | Recognize the inputs (explanations, entities, or humans) and provide the suitable outputs (numbers, descriptions, or entities) to answer the questions in a way that is understandable for non-native English speakers. | 68.00 |
| Subj | Manual Instruction | Please perform Subjectivity Classification task. Given the sentence, assign a label from ['subjective', 'objective']. Return label only without any other text. | 49.75 |
| | Natural Instruction | In this task, you are given sentences from reviews. The task is to classify a sentence as "subjective" if the opinion of the sentence is subjective or as "objective" if the opinion of the sentence is objective. | 52.55 |
| | **EVOPROMPT** | Construct input-output pairs to demonstrate the subjectivity of reviews and opinions, distinguishing between objective and subjective input while producing examples of personal opinions and illustrations of subjective views, so it can illustrate the subjectivity of judgments and perspectives. | 77.60 |

Table 14: Manual Instructions (following Zhang et al. (2023b) and Zhang et al. (2023c)), Natural Instructions (Mishra et al., 2022b), PromptSource (Bach et al., 2022) as baselines and instructions with best performance on Alpaca-7b generated by **EVOPROMPT** (either DE or GA) on classification datasets.

| Method | Model | Content | ROUGE-1/2/L |
|---|---|---|---|
| Manual Instruction | Alpaca-7b | How would you rephrase that in a few words? | 35.92/11.16/31.67 |
| | GPT | How would you rephrase that in a few words? | 43.95/17.11/39.09 |
| **EVOPROMPT** | Alpaca-7b | Carefully examine the text or listen to the conversation to identify the key ideas, comprehend the main idea, and summarize the critical facts and ideas in the concise language without any unnecessary details or duplication. | 39.86/14.24/36.09 |
| | GPT | Reduce the core by reading or listening carefully to identify the main ideas and key points, so readers can comprehend the important concepts and essential information. | 46.49/19.49/41.96 |

Table 15: Manual Instructions (following Sanh et al. (2021) as the baseline and instructions with best performance on Alpaca-7b and GPT3.5 generated by **EVOPROMPT** (either DE or GA) on SAMSum.

| Method | Model | Content | SARI |
|---|---|---|---|
| Manual Instruction | Alpaca-7b | Simplify the text. | 43.03 |
| | GPT-3.5 | Simplify the text. | 43.80 |
| **EVOPROMPT** | Alpaca-7b | Rewrite the input text into simple English to make it easier to comprehend for non-native English speakers. | 46.67 |
| | GPT-3.5 | Rewrite the given sentence to make it more accessible and understandable for both native and non-native English speakers. | 47.40 |

Table 16: Manual Instructions (following Zhang et al. (2023c) as the baseline and instructions with best performance on Alpaca-7b and GPT3.5 generated by **EVOPROMPT** (either DE or GA) on ASSET dataset.

| Task ID | Task | Description | Prompt | Score |
|---|---|---|---|---|
| 01 | hyperbaton | Order adjectives correctly in English sentences. | Verify the answer by splitting it into components and inspecting each part closely and logically, so we can progress thoughtfully and methodically as we break the task into pieces and explore each part systematically and rationally to reach our goal. | 81.20 |
| 02 | temporal_sequences | Answer questions about which times certain events could have occurred. | Start by breaking this conundrum into manageable chunks, carefully analyzing each component of this problem and thoroughly inspecting each aspect collaboratively, tackling it together progressively to ensure the correct answer and the desired outcome. | 78.80 |
| 03 | object_counting | Questions that involve enumerating objects and asking the model to count them. | Examine this logically and assess this methodically, so that we can obtain a precise result by thinking critically and dissecting this math task systematically. | 87.60 |
| 04 | disambiguation_qa | Clarify the meaning of sentences with ambiguous pronouns. | First, let us ponder and start off by taking our time, going step by step, and using our logic to approach this before we dive into the answer. | 71.20 |
| 05 | logical_deduction_three_objects | A logical deduction task which requires deducing the order of a sequence of objects. | Let's approach it cautiously, examining it thoroughly and methodically, and then approach it incrementally towards a resolution. | 94.40 |
| 05 | logical_deduction_five_objects | A logical deduction task which requires deducing the order of a sequence of objects. | Split the problem into steps and thoughtfully progress through them to find the answer after the proof. | 65.20 |
| 05 | logical_deduction_seven_objects | A logical deduction task which requires deducing the order of a sequence of objects. | Let's take a step-by-step approach to systematically dissect this math task. | 54.40 |

Table 17: Instructions with the best performance on GPT3.5 generated by **EVOPROMPT** (either DE or GA) on BBH datasets. Duplicate IDs are due to the tasks with several sub-tasks.

prompt optimization within the discrete language space lies in assessing the capacity of LLMs to adapt to these continuous control parameters.

- We hope our study can inspire further exploration of the connection between LLMs and other traditional algorithms, extending beyond EAs. The main challenge is adapting the specific elements of traditional algorithms to work within LLMs. For example, these elements may include direction of motion, velocity in partial swarm optimization (PSO) (Kennedy & Eberhart, 1995), the path in ant colony optimization algorithms (APO) (Dorigo & Gambardella, 1997), and characteristic in MAP-Elites (Mouret & Clune, 2015).

| Task ID | Task | Description | Prompt | Score |
|---|---|---|---|---|
| 06 | causal_judgement | Answer questions about causal attribution. | At first, let's handle things cautiously and resolve this by examining every detail and dealing with one problem at a time. | 65.78 |
| 07 | date_understanding | Infer the date from context. | Be realistic and practical like a detective, and use evidence to solve the problem in a logical, step-by-step approach. | 85.60 |
| 08 | ruin_names | Select the humorous edit that 'ruins' the input movie or musical artist name. | Break down a math task into smaller sections and solve each one. | 69.60 |
| 09 | word_sorting | Sort a list of words. | Analyze each part of the problem logically to solve it like a detective. | 56.40 |
| 10 | geometric_shapes | Name geometric shapes from their SVG paths. | We'll methodically work through this problem together. | 64.00 |
| 11 | movie_recommendation | Recommend movies similar to the given list of movies. | Before exploring the answer, | 86.00 |
| 12 | salient_translation_error_detection | Detect the type of error in an English translation of a German source sentence. | Break down the problem into individual steps in order to solve it. | 62.80 |
| 13 | formal_fallacies | Distinguish deductively valid arguments from formal fallacies. | Let's be realistic and evaluate the situation systematically, tackling it gradually. | 56.00 |
| 14 | penguins_in_a_table | Answer questions about a table of penguins and their attributes. | Let's start by taking a rational and organized approach, breaking it down into smaller parts and thinking it through logically, while being realistic and handling it carefully and methodically to ensure the right solution. | 84.25 |
| 15 | dyck_languages | Correctly close a Dyck-n word. | Let's be realistic and solve this challenge carefully and slowly, taking it slow to complete it correctly, so we can be realistic and cautiously reach the goal. | 44.40 |
| 16 | multistep_arithmetic_two | Solve multi-step arithmetic problems. | Before we dive into the answer, | 51.60 |
| 17 | navigate | Given a series of navigation instructions, determine whether one would end up back at the starting point. | Let's logically work together to systematically solve this math problem one step at a time in unison. | 94.20 |
| 18 | reasoning_about_colored_objects | Answer extremely simple questions about the colors of objects on a surface. | Using a detective's mindset, break down each element of this mathematical reasoning challenge one step at a time and reason like a detective to uncover the solution. | 88.00 |
| 19 | boolean_expressions | Evaluate the result of a random Boolean expression. | Let's gradually unravel this mathematical challenge by methodically addressing it by examining each element and investigating each factor. | 90.80 |
| 20 | tracking_shuffled_objects_three_objects | A task requiring determining the final positions of a set of objects given their initial positions and a description of a sequence of swaps. | Progress slowly and carefully through this mathematical reasoning challenge one step at a time. | 69.20 |
| 20 | tracking_shuffled_objects_five_objects | A task requiring determining the final positions of a set of objects given their initial positions and a description of a sequence of swaps. | Using a logical, step-by-step approach, work through this task to find the correct answer. | 81.20 |
| 20 | tracking_shuffled_objects_seven_objects | A task requiring determining the final positions of a set of objects given their initial positions and a description of a sequence of swaps. | Examine this issue logically and in detail, step-by-step, analyzing each part of the problem one at a time. | 84.80 |
| 21 | sports_understanding | Determine whether an artificially constructed sentence relating to sports is plausible or not. | Break down the problem into steps and start solving it. | 96.80 |
| 22 | snarks | Determine which of two sentences is sarcastic. | Break down and analyze each part of the problem in a step by step way to ensure the right answer is obtained. | 77.53 |

Table 18: Instructions with the best performance on GPT3.5 generated by EVOPROMPT (either DE or GA) on BBH datasets. Duplicate IDs are due to the tasks with several sub-tasks.

