# OpenReview forum: "Connecting Large Language Models with Evolutionary Algorithms Yields Powerful Prompt Optimizers"
_ICLR.cc/2024/Conference — ICLR 2024 poster_

### Official Review · Reviewer_hqfj · 2023-10-31

**Soundness:** 3 good
**Presentation:** 3 good
**Contribution:** 3 good
**Rating:** 6
**Confidence:** 2

**Summary:**

The paper proposes a novel framework for automatic discrete prompt optimization, called EVOPROMPT, which connects Large Language Models (LLMs) with Evolutionary Algorithms (EAs). The authors demonstrate that this approach can generate better prompts compared to human-engineered prompts and existing methods for automatic prompt generation. The paper is well-structured, with clear explanations and thorough experiments conducted on 31 datasets.

**Strengths:**

- The proposed EVOPROMPT framework is innovative, combining the language processing capabilities of LLMs with the optimization performance of EAs. This approach allows for the simultaneous leveraging of the powerful language processing capabilities of LLMs and the efficient optimization performance of EAs.

- The paper provides a comprehensive review of related works, including prompt engineering, discrete prompts, and LLMs with optimization algorithms. This review helps readers understand the context and contributions of the proposed method.

- The experiments conducted on 31 datasets demonstrate the effectiveness of EVOPROMPT compared to crafted prompts and existing methods. The results show consistent performance gains over both manual instructions and existing methods.

**Weaknesses:**

- The paper could benefit from more detailed explanations of the implementation details for each type of EA (GA and DE) used in EVOPROMPT. Providing more information on the specific steps and instructions for each algorithm would help readers better understand the proposed method and its connection to LLMs.

- The paper could include more ablation studies to better understand the impact of different components of the proposed method. For example, analyzing the effect of different population sizes, mutation rates, or selection strategies on the performance of EVOPROMPT would provide valuable insights into the method's robustness and generalizability.

**Questions:**

- Could you provide more detailed explanations of the implementation details for each type of EA (GA and DE) used in EVOPROMPT? Specifically, it would be helpful to understand how the algorithms are adapted to work with discrete prompts and how the LLMs are used to generate new candidate prompts.

- How is the effect of different population sizes, mutation rates, or selection strategies on the performance of EVOPROMPT

---

> ### Author Response · Authors · 2023-11-17
> **Author Response to Official Review by Reviewer hqfj**
>
> Dear Reviewer hqfj,
>
> Thank you for taking the time to review. We appreciate that you found our paper well-written and the literature review comprehensive.
> We would like to make the following clarifications about your concern. If you find these responses useful to address your questions and concerns, we would be grateful and would consider increasing your score.
>
> We summarize the mentioned concerns and we hope the corresponding comments address your concerns, we would be grateful if you could consider increasing the review score.
>
> > Q: More information on specific steps and instructions for each algorithm is needed.
>
> **A**: Thanks for reminding us of the specific steps, which is helpful for readers to better comprehend our methods. In our original paper, we unify the evolutionary algorithms in a framework. Considering your comments, we have updated our draft and added detailed algorithms for GA and DE, respectively (See Appendix A for details of implementation). We hope this can be responsive to your concerns. Additionally, we have attached the corresponding code in the supplementary materials. Later we will release our code to the public.
>
> > Q: How the algorithms are adapted to work with discrete prompts and how the LLMs are used to generate new candidate prompts?
>
> **A**: LLMs play a role in generating new prompts based on evolutionary operators including mutation and crossover according to the given candidates. Figure 1 and Figure 2 in the paper provide detailed instructions on guiding LLMs to generate a new discrete prompt. Meanwhile, we utilize the selection and updating strategy in evolutionary algorithms to control convergence and effectiveness.
>
> > Q: This paper could include more ablation studies to better understand the impact of different components of the proposed method, such as the effect of different population sizes, mutation rates, or selection strategies.
>
> **A:** Sec. 5.1 and 5.2 in our original paper have explored the effect of important components designed in GA and DE respectively. Particularly, for GA (Sec. 5.1), we mainly analyze the selection strategy including the random, tournament roulette wheel strategy. For DE (Sec. 5.2), we demonstrate the importance of mutation over different parts and the selection of Prompt 3 in Figure 2. Due to the space limitation, the study on population sizes is in Appendix C.1 of our paper.
>
> Besides, we would like to appreciate your suggestion about the mutation rates. Considering the LLMs may struggle with the rate of concrete float value since at the level of natural language, it is hard to measure the mutation extent, we leave this for future work.
>
> We also want to kindly remind you that the discussion phase will end soon, so we would greatly appreciate your support in this regard before the deadline.

---

> > ### Comment · Reviewer_hqfj · 2023-11-22
> >
> > Thank you for addressing my questions. After reading other reviews and responses, I do not have any more concerns or questions.

---

> ### Author Response · Authors · 2023-11-21
> **Reminder**
>
> Dear Reviewer hqfj,
>
> We greatly appreciate the time you've invested in reviewing our response. Having submitted our rebuttal, we are eager to know if you have any remaining concerns or require further clarification. Since the discussion phase will end soon, we would greatly appreciate your support and valuable feedback before the deadline **(Nov. 22, 2023)**.
>
> Best,
>
> Authors

---

### Official Review · Reviewer_HRoN · 2023-10-31

**Soundness:** 3 good
**Presentation:** 3 good
**Contribution:** 3 good
**Rating:** 6
**Confidence:** 3

**Summary:**

This paper introduces EvoPrompt, a new framework that leverages evolutionary algorithms (EAs) for discrete prompt optimization in Large Language Models (LLMs) without requiring access to the gradient or parameters of the LLMs. Specifically, taking cues from EA concepts such as mutation and crossover, EvoPrompt uses the LLM itself to carry out crossovers and mutations on a collection of pre-defined prompts, leading to the generation of a fresh set of evolved prompts. The EvoPrompt has been instantiated to both Genetic Algorithm (GA) and Differential Evolution (DE) versions. Comprehensive experiments across 31 datasets have demonstrated the efficacy and good performance of EvoPrompt when tested on both closed-source and open-source LLMs.

**Strengths:**

* The combination of LLMs with EAs as presented is novel to me, providing insights for future LLM research.
* I liked the design of using the LLM itself to iteratively refine candidate prompts, which harnesses the natural linguistic strengths of LLMs. This ensures that the generated prompts are coherent, readable, and well-constructed sentences.
* The authors have conducted extensive experiments on benchmark datasets and the discussions on different designs and hyperparameters are commendable.
* The paper is well-written and easy to follow. It offers useful examples and LLM templates.

**Weaknesses:**

* EvoPrompt's performance appears to be influenced by the quality of the initial prompts (initial population), which may limit its applications. While the paper emphasized the DE version's superior exploration over the GA, a deeper analysis would be beneficial.
* Further discussion and evaluation of the diversity of the prompts produced by EvoPrompt would be beneficial.
* The EA optimization for a single prompt involves iterations. However, the search doesn't appear to converge in Figure 6. Is it possible to illustrate the effects of more iterations? It's unclear if these 10 iterations are conducted within a single LLM session or if each iteration starts with a new session. Besides, more details should be provided regarding the runtime, and resources for running the GA.
* It would be beneficial if the authors could present negative examples where EvoPrompt failed, as this could inspire direction for future research. Moreover, the limitations of this approach should be discussed.

**Questions:**

1. In EvoPrompt, most of the initial populations were generated with a mix of manually-created and LLM-generated prompts. What is their ratio when DE works better than GA? What would be the performance of EvoPrompt if the initial prompts were all generated by LLM?
2. How effectively can EvoPrompt discover novel prompts utilizing EA operators? (e.g., considering factors such as prompt length and the variance in approaches between new and prior prompts).
3. When the LLM generates mutated new words, is there adequate diversity in the new words introduced, or is there a repetition of the same words?
4. See other questions in the weakness part.
5. Will there be any benefits to combining and alternating GA and DE updates during search?

---

> ### Author Response · Authors · 2023-11-17
> **Author Response to Official Review by Reviewer HRoN (1/2)**
>
> Dear Reviewer HRoN,
>
> Thank you for your positive and thoughtful feedback and look into every detail of our work. We appreciate that you found our paper well-written and experiments commendable. We summarize the mentioned concerns and we hope the corresponding comments address your concerns, and we would be grateful if you could consider increasing the review score.
>
> > Q: EvoPrompt’s performance may be influenced by initial prompts.
>
> A: In Sec 5.3, we investigate the effect of initial population quality. The results of random-k and top-k initialization exhibit a high degree of similarity and the random setting is more in line with the real situation. This suggests that EvoPrompt does not depend on carefully crafted manual prompts. Instead, even randomly selected prompts can achieve a good performance. Additionally, though DE (48.64) and GA (47.80) starting from the selected bottom population perform not that well, they still outperform APE (46.32), demonstrating the effectiveness and robustness of EvoPrompt.
>
> > Q: Deeper analysis for DE’s exploration over GA. \& Diversity of prompts
>
> A: We are very grateful for your insightful comments on comparing DE and GA from this perspective. Considering your suggestions, We mainly analyze prompt diversity through the following aspects: 1) the average length of the prompt and the corresponding variance in the population; 2) the number of new words after each iterative step; 3) the case study. We will briefly summarize the results in this response.
>
> * **Average length and variance**: From Figure 9 (a, b) in the Appendix, DE shows higher length and variance than GA. This implies that the prompts generated in DE are more diverse in terms of length.
>
> * **Number of new words**: From Figure 9 (c), we can observe that in the latter iterations, DE mutates more new words than GA, and thus shows better potential to explore and escape from the local optimum.
>
> * **Case study**: in Figure 1 (GA) and Figure 2 (DE), the given examples are taken from our experimental logs.
> **1)** For GA, in the final prompt (at the bottom), words in orange and blue are inherited from the parental prompts respectively, and words in black are mutated parts.
> **2)** For DE, comparing the final generated prompt (wrapped by a box at the bottom) with the basic prompt (in the upper part of the figure, prompt in blue words), the new prompt keeps half of the original one, while the other half derives from crossover and mutation referring to other candidates. DE keeps the common parts between the given candidates as repetition and only performs mutation on different parts. Both of them have repetitions and mutations, corresponding to exploitation and exploration.
>
> Results related to prompt diversity revealed that DE tends to generate more diverse prompts than GA. This result also indicates that DE is potentially better at exploration than GA. Please see Appendix C.4 in our revised version for more details. Thank you again for your insightful comments on comparing DE and GA from this perspective.
>
> > Q: The search doesn't appear to converge and it is possible to illustrate the effects of more iterations.
>
> **A**: We increase the iteration steps and there’s still slight performance gain on the Subj dataset, but EvoPrompt, both GA and DE versions, on most of the datasets converge around 10 steps. To maintain uniformity and reduce hyperparameter elaboration in the experimental setup, we selected a consistent value of 10 across all tasks.
> It's true that at the iteration of 10, the average score still shows an increasing trend but there exists a trade-off between the number of iterative steps and performance. If we continue evolving until convergence, the best score of DE can rise.
> Please refer to Appendix C.3 for more detailed demonstrations.
>
>
> > Q: Are 10 iterations conducted in a single LLM session or a distinct new session?
>
> A: Each evolution is in a distinct new session.

---

> ### Author Response · Authors · 2023-11-17
> **Author Response to Official Review by Reviewer HRoN (2/2)**
>
> Our responses to the rest of the questions are as follows.
>
> > Q: Details about the runtime and resources.
>
> A: Please refer to our general responses on cost analysis for details related to resources. The runtime varies from each dataset, the table below lists the runtime when optimizing for Alpaca on a single V100 GPU (except that BBH is optimized for GPT3.5). Additionally, the runtime of evaluation depends on iterations, the size of the population and the size of the dev set. Here we show the time under our default setting. The time for evaluation is almost the same for APE and our methods. Note that each task in BBH takes about the same amount of time to run.
>
> |                | SST-2 | CR   | MR   | SST-5 | AG's News | TREC | Subj | ASSET | BBH  |
> | -------------- | ----- | ---- | ---- | ----- | --------- | ---- | ---- | ----- | ---- |
> | Runtime (mins) | 63    | 63   | 65   | 90    | 162       | 99   | 194  | 163   | 86   |
>
> > Q: Negative examples where EvoPrompt fails and the limitations can lead to future investigation.
>
> A: In our results of Big-Bench Hard, there are several tasks that either EvoPrompt (GA) or EvoPrompt (DE) fails. This is because the distribution of the randomly sampled dev set is not aligned with the test set, resulting in a performance gap between the evaluations based on them. Accordingly, though EvoPrompt gets a better prompt based on the dev set, it may not perform as well when tested on the test set. Thus, constructing a more effective dev set could further enhance the performance, which needs future investigation.
>
> > Q: What is the ratio of manually created and LLM-generated prompts in the initial population when DE works better than GA? & What would be the performance of EvoPrompt if the initial prompts were all generated by LLM?
>
> A: There are two ratios of manually written prompts in the initial population, before top-k and after top-k and variations. Before selecting top-k, the ratio of manually written prompts is around 50\%. The ratio after top-k and variations are listed in the table below. We haven't observed a significant correlation between the ratio of manual prompts in the initial population and the performance.
>
> |                 | SST-2 | CR    | MR    | SST-5 | AG's News | TREC  | Subj  | ASSET |
> | --------------- | ----- | ----- | ----- | ----- | --------- | ----- | ----- | ----- |
> | Score (GA)      | 95.13 | 91.27 | 90.07 | 49.91 | 72.81     | 64.00 | 70.55 | 46.43 |
> | Score (DE)      | 94.75 | 91.40 | 90.22 | 49.89 | 73.82     | 63.73 | 75.55 | 46.21 |
> | Ratio of manual | 0.37  | 0.50  | 0.20  | 0.23  | 0.47      | 0.00  | 0.30  | 0.33  |
>
> Additionally, here we conduct experiments with the initial prompts generated by LLMs completely, as shown in the table below. We can observe that for SST-5, initialization has little impact.
>
> | Method             | SST-5 |
> | ------------------ | ----- |
> | APE                | 46.32 |
> | GA                 | 49.91 |
> | GA (-manual)| 49.98 |
> | DE                 | 49.89 |
> | DE (-manual) | 50.06 |
>
> > Q: Will there be any benefits to combining and alternating GA and DE updates during the search?
>
> A: Thanks for your comments. It’s a promising suggestion. The selection and updating strategy differ from DE and GA, making the combination non-trivial. We will further investigate this for our extensions.

---

> ### Author Response · Authors · 2023-11-21
> **Reminder**
>
> Dear Reviewer HRoN,
>
> We greatly appreciate the time you've invested in reviewing our response. Having submitted our rebuttal, we are eager to know if you have any remaining concerns or require further clarification. Since the discussion phase will end soon, we would greatly appreciate your support and valuable feedback before the deadline **(Nov. 22, 2023)**.
>
> Best,
>
> Authors

---

> > ### Comment · Reviewer_HRoN · 2023-11-22
> >
> > I appreciate the authors for the comprehensive response. Most of my concerns were addressed, although I still believe some more evaluation and analysis would have contributed to a deeper understanding of the proposed framework.  Nevertheless, the paper demonstrates novelty and promising performance, and I maintain my original score in support of its acceptance.

---

### Official Review · Reviewer_LYv4 · 2023-11-01

**Soundness:** 3 good
**Presentation:** 4 excellent
**Contribution:** 2 fair
**Rating:** 8
**Confidence:** 3

**Summary:**

In this paper, the authors present EvoPrompt, a framework to combine Evolution Strategies and a Language Model to engineer prompts to improve the performance of possibly another Language Model on a variety of instruction tasks. The authors mention that the prompt used to instruct an LLM strongly influences its performance and that human crafted prompts are often sub-optimal thus motivating the need to use optimization algorithms to generate prompts so as to maximize the performance of an LLM on a given task. In that context, the authors motivate the use of Evolution based algorithms as they typically perform great on discrete sequential data and offer a good way to balance exploration and exploitation. Evolution algorithms also have the benefit from performing black box optimization - they do not access to the function or its gradient - which allows users to use them with models that are only accessible through an API for instance. However, standard mutation and crossover functions  that typically invert / replace / mutate individual elements in the discrete sequences without maintaining global coherence which would lead these operators to quickly generate prompts that are unreadable and non interpretable. Therefore, the authors propose to use an LLM to perform crossover and mutation operations to always generate coherent and meaningful new samples. The authors compare to Evolution strategies - genetic algorithms and differential evolution - and benchmark their method on a variety of tasks including language understanding and summarization. The method is applied both to Alpaca (instruction-fine tuned Llama which is open-source) and to GPT-3.5 (proprietary model accessible through an API). They compare obtained to several baselines and propose some ablations to study the impact of mutation strategy used and the impact of the initial population.

**Strengths:**

First of all, I find the paper well written. The motivation is clearly explained, the context is well introduced and the authors do a great job in the literature review part. The method is also clearly explained and detailed which makes it easy for the reader to understand what is going on and which might help to reproduce the work as well. I also really enjoyed Figure 1 and 2 which are very clean, colorful and do a great job at illustrating the method.

The method itself is sound and I agree that Evolution Strategies make a lot of sense for this problem for all the reasons mentioned. I mainly like the way crossover and mutation is implemented by querying an LLM. I find this idea very powerful as indeed, in most cases the main challenge in using EAs lies in defining operators that can stay in the data distribution when perturbing samples.

I also find the experimental setup convincing as the authors performed a large number of experiments and compared to several relevant baselines. The fact that the method was illustrated both on an open-source and on a proprietary model also strengthened that part and I also like that the authors illustrated how to apply their method and LLM-based evo operator with two different types of Evolution Strategies.

**Weaknesses:**

While I really appreciate the evo operator that leverages LLMs, I have more doubts about the relevance of the considered application. I feel like there are more interesting applications for this method than prompt engineering which I believe will soon not be a thing anymore as we see the latest language models becoming more and more robust to the prompt that is used. However, I acknowledge that there are ongoing efforts in that field and while I personally don't find it very exciting or relevant, this work contributes efficiently to this research track.

I also have some doubts about the relevance of this work to ICLR. In my opinion, the main contribution of this work is this LLM-based mutation/crossover operator which would make more sense in an evolutionary algorithms conference such as GECCO than at ICLR.

Finally, I think that this work would strongly benefit from a study of the computational cost of the approach in the main core of the paper. I saw some elements that could help towards this end in the supplementary, but I would like to see an analysis of the computational cost / financial cost and time needed to perform this optimization. It would be interesting also to add similar analysis for the baselines. I think this would be very useful to the readers to evaluate to what extent they could apply that to their research if they have either to serve themselves Alpaca on an accelerator or to query the OpenAI API after connecting their credit card.

**Questions:**

All in all, I like the idea being introduced and I think the authors did a good job at introducing it and benchmarking it, therefore I would be happy to see this work published (it would be even better if the authors would demonstrate the method on a different use-case than prompt optimization however, I understand this would be a very different paper. It might be an idea for the next one though ;)). However, I have some doubts if ICLR is the best place for that as I think GECCO would be better suited. Therefore, I will say minor accept for now but I would be happy to increase my grade if the reviewers and other authors convince me that actually ICLR is the right place.


**A few other general comments**:

- As mentioned above, I think this would be very exciting to apply these LLM-based evo operators to other applications. For instance this could be used to generate game levels through textual description or coding exercises. With latest models such as DALLE-3, the same strategy could be applied to generate images.

- The authors mention that EA are good at balancing exploration and exploration, however it is not always the case and this is for instance often a limitation of GA methods. I would recommend the authors to have a glance at the Quality Diversity literature with algorithms such as MAP-Elites which are much better at balancing exploration. I think it would be very exciting to use EvoPrompt on top of a QD method.

- In the conclusion, the authors mention looking at Simulated Annealing (SA) for future work. In my experience, GA and other more recent EAs always outperform SA by a strong margin (if correctly tuned and implemented). Thus, I would not advise going in that correction. QD methods are probably going to be more fun. Looking at multi-objectives methods such as NSGA-II might be interesting as well.

---

> ### Author Response · Authors · 2023-11-17
> **Author Response to Official Review by Reviewer LYv4 (1/2)**
>
> Dear Reviewer LYv4,
>
> Thank you for taking the time to review and propose promising extensions. We appreciate that you found our paper well-written and sound with convincing results. If the following comments address your concerns, we would be grateful if you could consider increasing the review score.
>
> Firstly, we would like to clarify our **motivation for focusing on prompts**.
>
> * We kindly argue that prompts for large language models (LLMs) are crucial, as provide a necessary interface to interact with black-box LLMs. In the era of LLMs, the fine-tuning paradigm is computationally demanding, while the prompt in natural language is effective, interpretable and human-readable, especially practical for those limited by the computational budget. Previous works find prompt order [4], templates and examples can highly affect [5] the performance.
> The field of prompt engineering has recently gained significant popularity, evolving into a highly sought-after career. Automating this process, and achieving superior performance compared to human-designed prompts, signifies a major step forward in the area of language model optimization. This motivates us to study the automated generation of high-quality prompts.
>
> * We understand your concern that prompt engineering might not be useful as the language model scales. However, it has not been proved that the larger the model is, the more robust it is to the given prompt. In our experiments, even GPT3.5 is sensitive to prompts. Lu et al. [4] demonstrate that models of different scales are sensitive to the prompt, with the largest LM being 175 billion. Meanwhile, our proposed EvoPrompt also achieves significant improvement on multiple tasks (e.g., up to 25% improvement on BBH compared with the famous Chain-of-Thought, which shows the influence of different prompts). Moreover, as the model's scale increases, the corresponding rise in computational cost renders fine-tuning methodologies less feasible. This shift amplifies the importance and relevance of prompts, known for their effectiveness and better interpretability.
>
> * One major benefit of employing LLMs as evolutionary operators is their ability to take into account the coherence within a sequence. Consequently, the prompt in natural language is a particularly appropriate application. We begin with NLP tasks because they come with standard objective evaluation criteria, which are more effective in showcasing the performance of EvoPrompt. As you mentioned, it would indeed be intriguing to apply EvoPrompt to multi-modality applications (e.g., game level and image generation) by adjusting the evaluation and selection strategies.
>
> Secondly, **ICLR considers a broad range of subject areas.** One contribution of our work is to connect LLMs with evolutionary algorithms, where, as you mentioned, the most innovative design is the LLM-based mutation/crossover operator.
> This connection does not contribute to improving the evolutionary algorithms, instead, it sparks new applications for both of these fields. Accordingly, besides evolutionary algorithms, it is also highly related to LLMs. For example, the works considering evolutionary algorithms with reinforcement learning [2, 3]  were also accepted by ICLR’22 and ICLR’23. Another contribution is the prompt optimization framework. While our main focus is on prompts for NLP tasks, potential applications could also include prompts for vision tasks. This is in line with the "applications in language and vision" in the call for papers, as well as other previously accepted papers at ICLR'23 like APE [1].
>
> Finally, to address your **criticism regarding costs**, we analyze the cost to apply our EvoPrompt in general responses. Intuitively, there exists a trade-off between costs and the corresponding performance. EvoPrompt shows a higher ceiling compared with APE. Besides, EvoPrompt outperforms APE when keeping the same number of API requests. Please see the general response for more statistics.
>
> # References
> [1] Zhou, Yongchao, et al. "Large Language Models are Human-Level Prompt Engineers." *The Eleventh International Conference on Learning Representations*. 2023.
>
> [2] Wang, Yutong, Ke Xue, and Chao Qian. "Evolutionary diversity optimization with clustering-based selection for reinforcement learning." *International Conference on Learning Representations*. 2022.
>
> [3] Hao, Jianye, et al. "ERL-Re $^ 2$: Efficient Evolutionary Reinforcement Learning with Shared State Representation and Individual Policy Representation." *The Eleventh International Conference on Learning Representations*. 2023.
>
> [4] Lu, Yao, et al. "Fantastically Ordered Prompts and Where to Find Them: Overcoming Few-Shot Prompt Order Sensitivity." Proceedings of the 60th Annual Meeting of the Association for Computational Linguistics (Volume 1: Long Papers). 2022.
>
> [5] Zhao, Zihao, et al. "Calibrate before use: Improving few-shot performance of language models." International Conference on Machine Learning. PMLR, 2021.

---

> ### Author Response · Authors · 2023-11-17
> **Author Response to Official Review by Reviewer LYv4 (2/2)**
>
> We would like to appreciate your comments on our method we will consider these directions, including application in image generation, introducing QD methods, etc., as our further extensions. Specific responses to each of your questions are as follows:
>
> * **Applications of our method**: In this paper, we demonstrate the ability of LLMs to implement evolutionary algorithms given clear instructions and mainly aim at prompt optimization. In addition, our framework can be applied to various fields, such as non-trivial NP-hard problems (e.g. TSP), and multi-modality scenarios. It also can be deployed on agent-based systems to enhance cooperation among agents in diverse situations. We leave this for future exploration
> * **QD methods**: Thanks for this insightful suggestion and for reminding us of the limitation of SA compared with EAs. QD methods like MAP-Elites are more recent and seek to find a set of policies with both high rewards and diverse behaviors. These approaches show a promising ability to balance exploitation and exploration, which is crucial for heuristic searches. We will further investigate this for future work.
>
> Thank you again for your thoughtful feedback, and we hope our response has addressed your concerns adequately.

---

> ### Author Response · Authors · 2023-11-21
> **Reminder**
>
> Dear Reviewer LYv4,
>
> We greatly appreciate the time you've invested in reviewing our response. Having submitted our rebuttal, we are eager to know if you have any remaining concerns or require further clarification. Since the discussion phase will end soon, we would greatly appreciate your support and valuable feedback before the deadline **(Nov. 22, 2023)**.
>
> Best,
>
> Authors

---

> > ### Comment · Reviewer_LYv4 · 2023-11-21
> > **Convincing rebuttal**
> >
> > I warmly thank the authors for the work they did for that rebuttal. I find the explanations and new cost analysis convincing. I also agree with the authors that advertising the connection that can be made between ES and LLMs is of interest to ICLR attendees and I would like to re-iterate my appreciation of the overall quality of the work. I am happy to increase my score to accept.

---

### Official Review · Reviewer_ZLDk · 2023-11-05

**Soundness:** 2 fair
**Presentation:** 3 good
**Contribution:** 2 fair
**Rating:** 6
**Confidence:** 4

**Summary:**

The paper introduces EVOPROMPT, a new framework that merges evolutionary algorithms (EAs) with Large Language Models (LLMs) to optimize human-readable prompts for language tasks. By using EAs to iteratively improve prompts for both open- and closed-source LLMs such as GPT-3.5 and Alpaca, the study reveals significant performance enhancements (up to 25%) over human-designed prompts, showcasing the potential synergy between LLMs and EAs for prompt optimization.

**Strengths:**

The paper is well organized and clearly written.

**Weaknesses:**

The paper's attempt to introduce EVOPROMPT as a groundbreaking framework for discrete prompt optimization feels uninspired due to its reliance on established evolutionary algorithms (EAs) rather than innovating with novel approaches. While the study showcases the ability to optimize prompts for language models like GPT-3.5 and Alpaca, this contribution doesn't significantly advance the state of the art, particularly given the use of well-known evolutionary operators of DE and GA. Designing prompts for language models, although important, might not meet the level of substantial novelty expected at a conference like ICLR. This approach, while showing performance enhancements, might not present a substantial leap forward in the field of language model optimization, especially considering the expectations of innovation and original contributions at such academic conferences. I think this paper would be better suited in a conference like IEEE CEC or PPSN.

**Questions:**

Why out of so many available methods, you only chose DE and GA? Why you used only one particular offspring generation strategy of DE? How are you repairing infeasible set of offspring and how does that process affect your timing complexity?

The authors have provided extensive rebuttals and based on the discussions with the authors, I am revising my score to 6.

---

> ### Author Response · Authors · 2023-11-17
> **Author Response to Official Review by Reviewer ZLDk (1/2)**
>
> Dear Reviewer ZLDk,
>
> We are grateful for your careful review and the valuable feedback that you provided for our paper. We would like to make the following clarifications about your concern. We also want to remind you that the discussion phase will end soon, so we would greatly appreciate your support in this regard before the deadline.
>
> Firstly, we would like to clarify the motivation and novelty of using the transitional evolutionary algorithms. The main novelty of this work is to propose a general framework that connects LLMs with evolutionary algorithms, as well as the LLM-based evolutionary operator design, which is also highlighted by all the other reviewers (LYv4, HRoN, hqfj). Furthermore, we believe that many traditional algorithms are still attractive due to their good performance, and connecting them with LLMs may encourage more innovative applications and not be limited to prompt optimization. Besides, many excellent works combine traditional well-performed algorithms with LLMs to get powerful methods. For instance, Self-Consistency [6] combines ensemble learning and LLMs (ICLR'23). [7] introduces effective beam search into reasoning paths (NeurIPS'23).
>
> Secondly, we appreciate your concern that our contribution doesn't significantly advance the state of the art. As shown in the experiments, EvoPrompt shows significant improvement on multiple tasks (e.g., up to 25% improvement on BBH compared with the famous Chain-of-Thought) and Reviewer LYv4, HRoN, hqfj find our experimental results convincing.
>
> Finally, we appreciate your concern regarding the novelty and appropriateness of prompt design for ICLR.
>
> * Prompts provide a necessary and efficient interface to interact with black-box LLMs. Previous works find prompt order [8], templates and examples can highly affect [9] the performance. The field of prompt engineering has recently gained significant popularity, evolving into a highly sought-after career. Automating this process, and achieving superior performance compared to human-designed prompts, is challenging and also signifies a major step forward in the area of language model optimization.
> we have observed many good works related to prompts presented at previous ICLR, as well as other top conferences. For example, the famous Chain-of-Thought [1] has been accepted by NeurIPS'22. APE (one of our strong baselines [2]), GenRead [3], TEMPERA [4], least-to-most prompting [5] have been accepted by ICLR’23. Thus we believe that prompt design contributes to the advancement of the field and is in line with other previously accepted papers at ICLR'23.
>
> * On the other hand, the contribution of this work extends beyond prompt optimization. It also establishes a novel connection between evolutionary algorithms and LLMs. This framework can also be applied to various tasks, such as non-trivial NP-hard problems. Furthermore, we hope our work can inspire future research at the intersection of LLMs and traditional algorithms, encouraging innovative applications.
>
> # References
> [1] Wei, Jason, et al. "Chain-of-thought prompting elicits reasoning in large language models." *Advances in Neural Information Processing Systems*. 2022.
>
> [2] Zhou, Yongchao, et al. "Large Language Models are Human-Level Prompt Engineers." *The Eleventh International Conference on Learning Representations*. 2023.
>
> [3] Yu, Wenhao, et al. "Generate rather than Retrieve: Large Language Models are Strong Context Generators." *The Eleventh International Conference on Learning Representations*. 2023.
>
> [4] Zhang, Tianjun, et al. "Tempera: Test-time prompt editing via reinforcement learning." *The Eleventh International Conference on Learning Representations*. 2023.
>
> [5] Zhou, Denny, et al. "Least-to-most prompting enables complex reasoning in large language models." *The Eleventh International Conference on Learning Representations*. 2023.
>
> [6] Wang, Xuezhi, et al. "Self-Consistency Improves Chain of Thought Reasoning in Language Models."  *The Eleventh International Conference on Learning Representations*. 2023.
>
> [7] Xie, Yuxi, et al. "Decomposition enhances reasoning via self-evaluation guided decoding." *Advances in Neural Information Processing Systems*. 2023.
>
> [8] Lu, Yao, et al. "Fantastically Ordered Prompts and Where to Find Them: Overcoming Few-Shot Prompt Order Sensitivity." Proceedings of the 60th Annual Meeting of the Association for Computational Linguistics (Volume 1: Long Papers). 2022.
>
> [9] Zhao, Zihao, et al. "Calibrate before use: Improving few-shot performance of language models." International Conference on Machine Learning. PMLR, 2021.

---

> ### Author Response · Authors · 2023-11-17
> **Author Response to Official Review by Reviewer ZLDk (2/2)**
>
> Specific responses to each of your questions are as follows:
> - **The reason for choosing GA and DE**: We choose representative and widely used evolutionary algorithms, GA and DE for their fast convergence without any parameters.
> GA is among the most highly regarded evolutionary algorithms and DE has emerged as one of the most widely utilized algorithms for complex optimization challenges since its inception (first two paragraphs in Section 3). Apart from these two representative algorithms, other diverse algorithms can be further explored.
> - **Offspring generation strategy of DE**: Our original paper has carefully investigated different generation strategies of DE (analysis in Sec 5.2). In the main experiments, we choose one of them with the best performance.
> - **Repairing infeasible offspring and effect on time complexity**: For both GA and DE, after each iteration including selection, mutation and crossover, updating upon the current population, the infeasible offspring will be eliminated. The elimination is up to their fitness value, i.e., the performance on the given development set in our work. Eliminating infeasible offspring based on certain rules or a small subset of the development set, rather than evaluating them on the entire development set, could potentially reduce costs and leave them for future investigation.

---

> ### Author Response · Authors · 2023-11-21
> **Reminder**
>
> Dear Reviewer ZLDk,
>
> We greatly appreciate the time you've invested in reviewing our response. Having submitted our rebuttal, we are eager to know if you have any remaining concerns or require further clarification. Since the discussion phase will end soon, we would greatly appreciate your support and valuable feedback before the deadline **(Nov. 22, 2023)**.
>
> Best,
>
> Authors

---

> ### Comment · Reviewer_ZLDk · 2023-11-21
> **Response to the Authors' Rebuttals**
>
> I apologize for posting late on the authors' response. I will rethink of my score, after receiving some response for the following questions.
>
> While I do appreciate the extensive attempts made by the authors for analyzing the cost function further and elaborate on the efficacy of their transitional EA based approaches, the questions still remains are: did you try the best and latest version of DE like SHADE or quivalent, as covered in a more latest survey (https://doi.org/10.1016/j.swevo.2016.01.004) ? If not, why?
>
> Also, given the inherently discrete nature of the prompt optimization problem, why didn't you try binary PSO or the ant colony optimization type algorithms? WIll not doing round-off for the solutions returned by inherently continuous operators of DE introduce more quantization noise?

---

> > ### Author Response · Authors · 2023-11-22
> > **Author Response to Reviewer ZLDk**
> >
> > Dear Reviewer ZLDk,
> >
> > Thanks a lot for your engagement and comments! We summarize the mentioned questions and we hope the corresponding comments address your concerns, and we would be grateful if you could consider increasing the review score.
> >
> > > Q: Did you try the best and latest version of DE like SHADE or equivalent, as covered in a more latest survey (https://doi.org/10.1016/j.swevo.2016.01.004)? If not, why?
> >
> > **A:** There indeed exists plenty of variants of DE and we give priority to most canonical and classical ones for current exploration. More effort will be needed in future investigations. In the case of the mentioned SHADE or similar variants that investigate adaptive control parameters, transferring them to prompt optimization in the discrete language space poses a non-trivial challenge. This is because the ability of the language model to follow such continuous control parameters requires further exploration.
> >
> > > Q: Given the inherently discrete nature of the prompt optimization problem, why didn't you try binary PSO or the ant colony optimization type algorithms? Will not doing round-off for the solutions returned by inherently continuous operators of DE introduce more quantization noise?
> >
> > **A:** In this paper, we focus on the evolutionary algorithms as the mutation and crossover operators initially designed for gene sequences are well-suited for prompts in natural language. For particle swarm optimization and ant colony algorithms, how to define the direction of motion, velocity, etc. on discrete languages needs to be further explored. Moreover, since natural language demands a high level of coherence, achieving complete alignment with traditional algorithms poses a challenge. It is essential to make appropriate adaptations and adjustments, whether these traditional algorithms are handling continuous or discrete vectors.
> >
> >
> > Best,
> >
> > Authors

---

> > > ### Comment · Reviewer_ZLDk · 2023-11-22
> > > **Reply to further comments from authors**
> > >
> > > Given your extensive response and attempts, I would increase my score to 6, tending towards acceptence.
> > >
> > > "For particle swarm optimization and ant colony algorithms, how to define the direction of motion, velocity," - please note that ACO, unlike PSO, doesn't have velocity direction etc. It has been extensively used for discrete and combinatorial optimization problems. You should discuss such issues in the final version. You should also refer to more recent surveys on DE and justify why you are using the basic variants without modern parameter (F and Cr) adaptation techniques.

---

> > > > ### Author Response · Authors · 2023-11-23
> > > > **Response to Reviewer ZLDk**
> > > >
> > > > Dear Reviewer ZLDk,
> > > >
> > > > We would like to thank the reviewer's positive feedback and for increasing the score. We do really appreciate the reviewer's time and effort in the reviewing process. We'll update the discussion about the DE variants and other evolutionary algorithms including ACO, PSO, etc. in our next revision.
> > > >
> > > > Best,
> > > >
> > > > Authors

---

### Public Comment · ~Yujian_Betterest_Li1 · 2023-11-16

Hi, happy to meet your good work.

We have a similar work ([SPELL: Semantic Prompt Evolution based on a LLM](https://arxiv.org/abs/2310.01260)) for prompt optimisation.

In the experiments, we find that, with LLM as a generator and a RoBERTa as the target model, the optimisation process is quite unstable. A good result is only obtained after several repeated runs.

We think the reason may be that the generated prompts are not consistently good for the target model due to the differences between the generation model and the target model. Hence, the problem is supposed to be eased when the target model is a large language model as well.

So, we're curious whether yours are stable.

Looking forward to your reply🤯

Thanks.

---

> ### Author Response · Authors · 2023-11-17
> **Author Response to Public Comment by Yujian Betterest Li**
>
> Dear Yujian,
>
> Thank you for taking the time to share your feedback and results related to our work.
>
> We'd like to highlight that our proposed EvoPrompt is stable. This has been demonstrated in our experiments, where we show the average performance over three runs on Alpaca, along with the standard deviation. As for GPT-3.5, due to budget limitations, we only conducted a single run.
>
> For the unstable issue in your case, we concur that it should be eased when the target model is also a large language model. This is due to the fact that RoBERTa, being relatively small, may not effectively comprehend and follow the instructions produced by LLMs.

---

### Author Response · Authors · 2023-11-17
**General Comments to All Reviewers**

# Summary
We would like to thank all the reviewers for their time and effort in the review process. We appreciate that you found our work  “novel and sound” (LYv4, HRoN, hqfj), “well-written and clear” (ZLDk, LYv4, HRoN, hqfj), “innovative and effective” (hqfj), our experiments “convincing/commendable/thorough” (LYv4, HRoN, hqfj) compared with baselines, and our literature review comprehensive (LYv4, hqfj). Taking into account all the reviewers’ comments, we’ve responded to each reviewer individually, uploaded a revised draft, and collected the below responses to general concerns. If you find our answers responsive to your concerns, we would be grateful if you would consider increasing your score.

Several reviewers asked for cost analysis. In the following, we first list what has changed since our original submission, and then analyze our costs for applications.

## Revisions
* [New Analysis for Prompt Diversity]: We add an analysis of prompt diversity to investigate GA and DE in generating instruction, mainly including the prompt length, variance and new words mutated after each iteration in Appendix C.4. \@HRoN
* [New Analysis for Cost]: We add analysis for the cost of our methods compared with APE in Appendix C.3. \@LYv4, \@HRoN
* [Clarity]: We add details of algorithms for GA and DE for clarifications, respectively, in Appendix A. \@hqfj



## Cost Analysis
The overhead of our methods mainly comes from prompt evaluation and generation. For evaluation, our overhead is $N * |D| * T$, where $N$ is the size of the population, $|D|$ is the size of the development set, and $T$ is the number of iterations.
These parameters differ from the task and can be found in Appendix B.3.
For the cost from prompt generation, the cost mainly depends on the number of API results, $T * N$. So the total number of API requests is $N * T * (1 + |D|)$, the same as APE.
Moreover, given that the API of LLMs is typically billed based on the number of tokens used, we also estimate the total number of tokens used in the API requests during the prompt optimization process.

We analyze the overhead mainly from two aspects: 1) the performance of our methods compared with APE under the same number of iterations; 2) the performance until convergence measured by the average score on the dev set. We report the scores on the test set, as shown below.
We can observe that with the same number of iterations, both GA and DE outperform APE significantly while introducing only a slight overhead in terms of the number of tokens. The convergence rates of APE and GA are similar while DE is slightly slower, but it delivers better performance. This implies a relatively high ceiling for our methods.

|                     | SST-5       |         |         | Subj       |         |         |
|---------------------|-------------|---------|---------|------------|---------|---------|
|                     | APE         | GA      | DE      | APE        | GA      | DE      |
|---------------------|-------------|---------|---------|------------|---------|---------|
| **Same iteration**  |             |         |         |            |         |         |
| # iterations        | 9           | 9       | 9       | 15         | 15      | 15      |
| # tokens            | 5.39 M      | 5.40 M  | 5.52 M  | 5.66 M     | 5.73 M  | 5.93 M  |
| score               | 45.79       | 50.23   | 49.23   | 67.20      | 70.10   | 79.35   |
|---------------------|-------------|---------|---------|------------|---------|---------|
| **Until convergence** |           |         |         |            |         |         |
| # iterations        | 9           | 7       | 11      | 15         | 15      | 17      |
| # tokens            | 5.39 M      | 4.20 M  | 6.75 M  | 5.66 M     | 5.73 M  | 6.72 M  |
| score               | 45.79       | 50.23   | 51.13   | 67.20      | 70.10   | 79.35   |

---

### Public Comment · ~Rui_Pan4 · 2023-11-23

Hi, happy to meet your good work.

We also have similar work [Plum: plum: prompt learning using metaheuristic](https://arxiv.org/abs/2311.08364) for prompt optimization, which has only been released recently due to anonymity periods.

In our paper, we employed metaheuristics, a superset of evolutionary algorithms, and implemented 5 algorithms other than standard GA and DE. There are algorithms such as SA (simulated annealing), HS (harmony search), TS (tabu search), and GA-M (GA without crossover) which are algorithmically simpler than GA. HS are even more efficient and superior than GA in certain tasks.

We are wondering if the authors could make some comparisons with these algorithms, or add some comments in the later version of the paper (our implementation is available at https://github.com/research4pan/Plum).

We understand that adding LLM operator in the search process is important, but these results illustrate that the algorithm choice is also important. Any comments would greatly appreciated!

Looking forward to your reply.

Thanks.

---

### Meta-Review · Area_Chair_d7bK · 2023-12-06

**Metareview:**

This paper adapts basic evolutionary optimization algorithms to the task of prompt optimization, showing that evolving prompts against a development set can lead to improved test performance across many datasets, including BigBench-Hard tasks.

The approach is simple, and there is not much novelty in the approach (it is a straightforward application of basic evolutionary algorithms to prompt design, making use of ideas from the ELM paper by Lehman et al, 2022). However, the results are compelling, and this paper serves as a useful proof-of-concept that evolving prompts can indeed result in the discovery of prompts outperforming previous SOTA methods like CoT.

Some suggestions for how the paper can be further improved:
- Incorporate some form of manual-design by humans baseline to further elucidate the benefits of this method.
- While the authors study the sensitivity of their method to the quality of the initial population, it would be useful to know how much a priori domain knowledge is necessary. This could be investigated by looking an initial population of randomly generated instructions, generated independently of the task at hand.

**Justification For Why Not Higher Score:**

This paper provides a useful proof-of-concept of an idea that should be expected to work: directly optimizing LLM prompts for a target task domain should yield prompts producing improved performance on that task domain. The specific algorithmic implementation details and experiment results in this work will be useful for future research to build on. However, the result itself is not surprising or exceptional, because it is an idea that is expected to work.

**Justification For Why Not Lower Score:**

As noted in the previous comment, the results and implementation details here are useful for future work to build on. This work validates the effectiveness of evolutionary search for prompt optimization.

---

### Decision · Program_Chairs · 2024-01-16

Accept (poster)